# Properties of rabies virus phosphoprotein and nucleoprotein biocondensates formed in vitro and in cellulo

**Quentin Nevers, Nathalie Scrima[☉], Damien Glon[☉], Romain Le Bars, Alice Decombe[¤], Nathalie Garnier, Malika Ouldali, Cécile Lagaudrière-Gesbert, Danielle Blondel, Aurélie Albertini, Yves Gaudin[iD]\***

Institute for Integrative Biology of the Cell (I2BC), CEA, CNRS, Université Paris-Saclay, Gif-sur-Yvette, France

☉ These authors contributed equally to this work.
¤ Current address: Architecture et Fonction des Macromolécules Biologiques, Centre National de la Recherche Scientifique, Aix-Marseille Université, Marseille, France
\* yves.gaudin@i2bc.paris-saclay.fr

**Data Availability Statement:** All relevant data are within the manuscript and its Supporting Information files.

## Abstract

Rabies virus (RABV) transcription and replication take place within viral factories having liquid properties, called Negri bodies (NBs), that are formed by liquid-liquid phase separation (LLPS). The co-expression of RABV nucleoprotein (N) and phosphoprotein (P) in mammalian cells is sufficient to induce the formation of cytoplasmic biocondensates having properties that are like those of NBs. This cellular minimal system was previously used to identify P domains that are essential for biocondensates formation. Here, we constructed fluorescent versions of N and analyzed by FRAP their dynamics inside the biocondensates formed in this minimal system as well as in NBs of RABV-infected cells using FRAP. The behavior of N appears to be different of P as there was no fluorescence recovery of N proteins after photobleaching. We also identified arginine residues as well as two exposed loops of N involved in condensates formation. Corresponding N mutants exhibited distinct phenotypes in infected cells ranging from co-localization with NBs to exclusion from them associated with a dominant-negative effect on infection. We also demonstrated that *in vitro*, in crowded environments, purified P as well as purified N0-P complex (in which N is RNA-free) form liquid condensates. We identified P domains required for LLPS in this acellular system. P condensates were shown to associate with liposomes, concentrate RNA, and undergo a liquid-gel transition upon ageing. Conversely, N0-P droplets were disrupted upon incubation with RNA. Taken together, our data emphasize the central role of P in NBs formation and reveal some physicochemical features of P and N0-P droplets relevant for explaining NBs properties such as their envelopment by cellular membranes at late stages of infection and nucleocapsids ejections from the viral factories.

## Author summary

RABV is a neurotropic virus that causes fatal encephalitis in humans and animals. There is no antiviral molecule active against the disease. RABV transcription and replication

**Funding:** This work was supported by the « Prix Bettencourt Coups d'élan pour la recherche française » attributed to YG, grants from the Agence Nationale de la Recherche, France, (ANR CE11, LiquidFact) to Y.G. (including a post-doctoral grant to Q.N.) and a grant from the Fondation pour la Recherche Médicale, France, to Y.G. (including a post-doctoral grant to D.G.). This work has benefited from the core facilities of Imagerie-Gif led by RLB, supported by "France-BioImaging" (ANR-10-INBS-04-01). The funders had no role in study design, data collection and analysis, decision to publish, or preparation of the manuscript.

**Competing interests:** The authors have declared that no competing interests exist.

take place within viral factories, called Negri bodies (NBs). The recent discovery that NBs have liquid properties and are formed by liquid-liquid phase separation raises several questions about the organization and properties of those compartments. To decipher the molecular bases underlying such processes, it is necessary to design versatile and easy-to-manipulate minimal systems that recapitulate the characteristics of those viral condensates. Here, we describe and compare two minimal systems, a cellular and an acellular one, that mimic NB properties. This highlights the key role of RABV phosphoprotein in these processes and identifies several physico-chemical principles underlying the properties of viral factories. These minimal systems will be useful to finely characterize the weak interactions between proteins involved in NBs formation using biophysical approaches.

## Introduction

Rabies virus (RABV, *Mononegavirales* order, *Rhabdoviridae* family, Lyssavirus genus) is a neurotropic virus that causes fatal encephalitis in humans and animals. Rabies disease still kills more than 55,000 people worldwide every year, mainly in Asia and Africa [1].

RABV genome is a single RNA molecule of negative polarity (~12kb). It is encapsidated by the nucleoprotein (N) to form a helical nucleocapsid in which each N protomer binds to nine nucleotides [2]. The nucleocapsid is associated with the RNA-dependent RNA polymerase (L) and its cofactor the phosphoprotein (P) to form the ribonucleoprotein (RNP). Once released in the cytoplasm, the RNP serves as a template for transcription and replication processes that are catalyzed by the L–P polymerase complex. During transcription, a positive-sense leader RNA and five capped and polyadenylated mRNAs are synthesized. The replication process yields nucleocapsids containing full-length antigenome-sense RNA, which in turn serve as templates for the synthesis of genome-sense RNA. During the replication stage, when not bound to the viral genomic or antigenomic RNA, N is kept soluble by binding the N-terminal region of P thus forming the so-called N0-P complex [3–5]. RABV transcription and replication take place within viral factories, the so-called Negri bodies (NBs) [6]. NBs are viro-induced cytoplasmic inclusions, having a diameter up to a few micrometers and containing all the replication machinery (L, N and P) [6]. It has been demonstrated that they have liquid organelles properties and are formed by liquid-liquid phase separation (LLPS) [7].

Membrane-less liquid organelles contribute to the compartmentalization of the eukaryotic cell interior. They are referred to as droplet organelles, proteinaceous membrane-less organelles or biomolecular condensates and are involved in a wide range of cell processes [8–12]. They are very dynamic structures, extremely sensitive to their physicochemical environment. Finally, they are highly enriched in some proteins that are much more concentrated in those structures than in the cytosol, as a result of LLPS [8,12].

The liquid nature of other *Mononegavirales* viral factories was further confirmed for vesicular stomatitis virus (VSV) [13], measles virus (MeV) [14,15] and respiratory syncytial virus (RSV) [15,16], suggesting that LLPS is widely exploited by members of this viral order during infection [17].

At some point, neosynthesized RNPs must leave the viral factory to form new virions. For RABV, two distinct processes, of which the molecular bases are unknown, seem to be at work. First, RNPs are ejected from the NBs and transported across the cytoplasm using the microtubule network [7]. Second, in the late stages of infection, NBs become enveloped by a double membrane, apparently derived from rough ER [6,7]. They lose their spherical shape [7] and

                                                  

virions are observed, budding from NBs into the compartment delimited by the associated double membrane [6,18].

The co-expression of RABV N and P proteins in mammalian cells is sufficient to induce the formation of cytoplasmic biocondensates having properties that are like those of NBs [7]. This cellular minimal system was used to identify P domains that are essential for biocondensates formation. P is a modular protein [5,19–21] and the dimerization domain (DD), the second intrinsically disordered domain (IDD2) and the C-terminal domain (CTD), which binds to the N associated with RNA, appear to be required for NB-like structures formation [7]. On the other hand, the first intrinsically disordered domain (IDD1), and the N-terminal part of the protein (NTD), involved in the formation of the N0-P complex and in the interaction with the L protein, appear to be dispensable [7].

In this work, we characterized the mechanisms of RABV P- and N-induced LLPS. First, we constructed fluorescent versions of N and analyzed their dynamics inside the biocondensates formed in the cellular minimal systems as well as in NBs of RABV-infected cells using FRAP. We identified several arginine residues of N involved in RNA binding, as well as two flexible loops, that play a key role in N-P biocondensates formation in the cellular minimal system. The corresponding N mutants were also analyzed in RABV-infected cells revealing distinct phenotypes ranging from co-localization with NBs to exclusion from them associated with a dominant-negative effect on their formation. Second, we demonstrated that *in vitro*, in crowded environments, purified P protein alone triggers LLPS. The domains of P required for phase separation *in vitro* are the same as those identified in the cellular minimal system. P droplets appear to concentrate RNA molecules, exclude some proteins, and associate with liposomes. Finally, *in vitro*, in crowded environments, purified N0-P complexes also form liquid droplets, which are disrupted when incubated with RNA molecules. Taken together, our data emphasize the central role of P in NBs formation. They also reveal some physicochemical features of P and N0-P droplets that constitute the molecular bases of morphogenesis, organization and properties of NBs.

## Results

### Functional characterization of chimeras between N and GFP in the cellular minimal system

Co-expression of RABV N and P proteins in BSR-T7/5 cells is sufficient to induce the formation of cytoplasmic biocondensates having properties that are similar to those of NBs [7]. The dynamics of the condensates formed in this cellular minimal system has been previously characterized with a fluorescent version of P (P-mCherry) and P domains required for LLPS have also been identified [7].

We investigated more thoroughly the role and the behavior of RABV N in this minimal cellular system. For this, we constructed several fluorescent versions of N in the pTit plasmid. In the first construct, referred to as GFP-N, N was amino-terminally fused with the GFP through a GGGGSGGGGS linker. In the other constructs (N-106GFP107, N-112GFP113, N-128GFP129, and N-130GFP131), the GFP was inserted into or nearby a flexible loop (L1) of N extending from residues 104 to 117. N segment [106–131] is solvent-accessible and is not involved in intermolecular contacts between N protomers in the N-RNA crystal structure [2] (Fig 1A). In those four constructs, the GFP insertion was flanked by two GGGGSGGGGS flexible linkers.

Those fusion proteins were all expressed in BSR-T7/5 cells after transfection with the corresponding plasmids as shown by western-blot (S1A Fig) and immunofluorescence (S1B Fig). Their ability to form biocondensates with P protein was then evaluated in the minimal system.

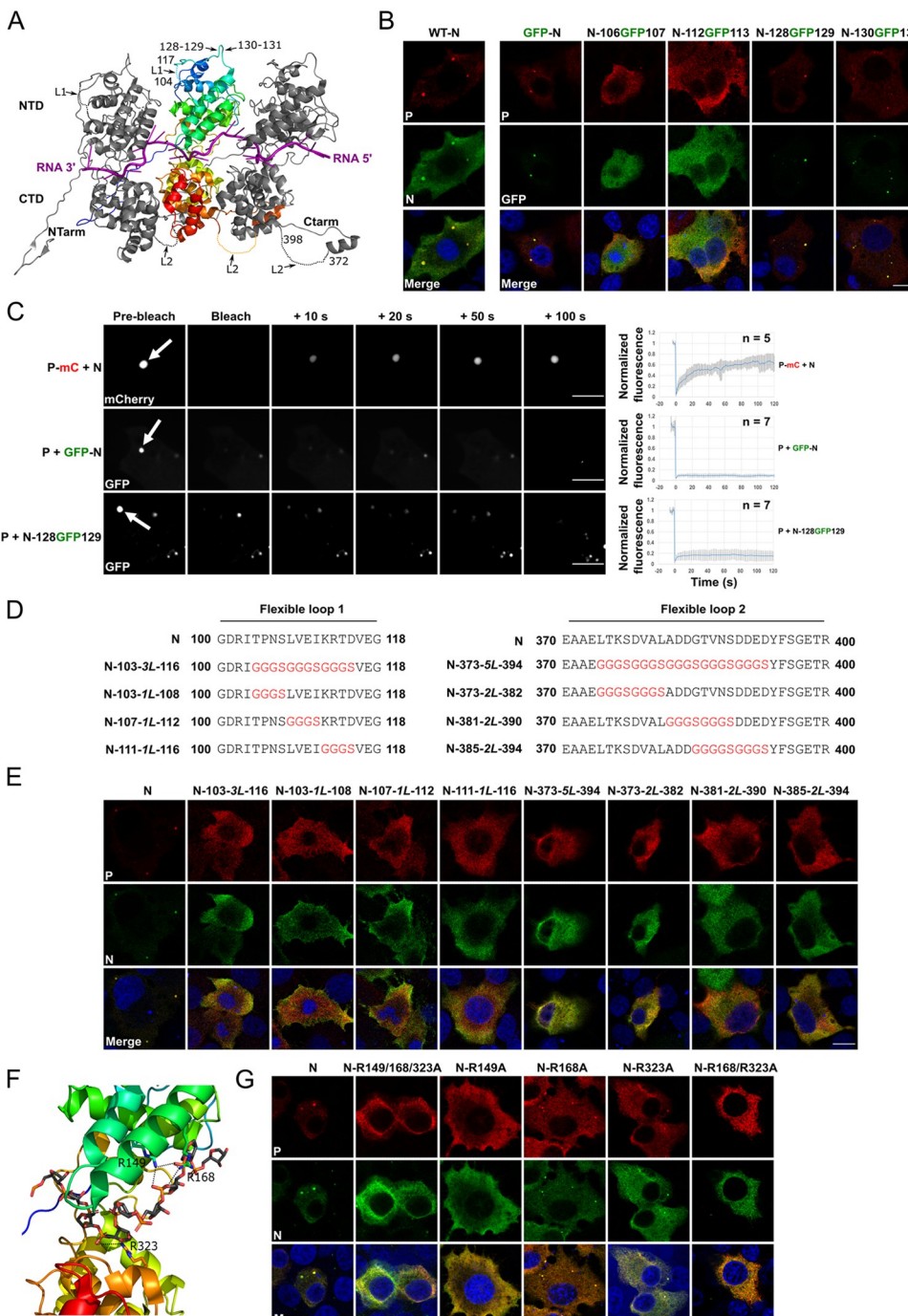

**Fig 1. Structural elements of RABV nucleoprotein required for the formation of biomolecular condensates in a cellular minimal system. A)** Ribbon representation of the crystal structure of RABV nucleoprotein (PDB 2GTT). Three protomers are shown in complex with RNA (in magenta). The RNA molecule is located in a groove between N amino-terminal domain (NTD) and carboxyterminal domain (CTD). Both domains are extended by two elongated arms (NTarm and CTarm). The central protomer is rainbow colored from N (blue) to C (red) terminus. The loops L1 and L2, absent in the crystal structure, are represented by dotted lines and their position indicated by arrows. The insertion sites between residues 128–129 and 130–131 are also indicated by arrows. **B)** BSR-T7/5 cells were co-transfected with pTit plasmids encoding P protein and the indicated fluorescent (GFP-N, N-106GFP107, N-112GFP113, N-128GFP129 and N-130GFP131) or non fluorescent N constructions. Cells were fixed 24 h post transfection and the formation of N-P inclusions was assessed by fluorescence confocal microscopy. P was revealed with a rabbit polyclonal anti-P antibody followed by incubation with Alexa-568 goat anti-rabbit IgG. Scale bar: 10 µm.

**C)** N/P-mCherry inclusions as well as GFP-N/P, and N-128GFP129/P inclusions (observed in BSR-T7/5 cells co-transfected as in B) were photobleached and the recovery of fluorescence was imaged with a Spinning-Disk microscope. Scale bars: 5 µm. For the plots on the right, FRAP data were corrected for background, normalized to the minimum and maximum intensity. The mean is shown with error bars representing the SD. **D)** Sequence of N mutants in which one or several stretches of amino acids in loops L1 and L2 have been replaced by one or several GGGS linkers (*L*). **E)** BSR-T7/5 cells were co-transfected with pTit plasmids encoding the indicated N mutant (mutated in L1 and L2) and the P protein. Cells were fixed 24 h post transfection and the formation of N-P inclusions was assessed by fluorescence confocal microscopy. N was revealed with a mouse monoclonal anti-N antibody followed by incubation with Alexa-488 goat anti-mouse IgG antibody and P was revealed with a rabbit polyclonal anti-P antibody followed by incubation with Alexa-568 donkey anti-rabbit IgG. Scale bar: 10 µm. **F)** View of the RNA binding groove of the crystal structure of RABV N. Arginines pointing toward the phosphodiester backbone (in sticks representation) are indicated. **G)** BSR-T7/5 cells were co-transfected with pTit plasmids encoding the indicated N mutant and the P protein. Cells were fixed 24 h post transfection and the formation of N-P inclusions was assessed by fluorescence confocal microscopy. N was revealed with a mouse monoclonal anti-N antibody followed by incubation with Alexa-488 goat anti-mouse IgG and P was revealed with a rabbit polyclonal anti-P antibody followed by incubation with Alexa-568 donkey anti-rabbit IgG. Scale bar: 10 µm.

Condensates were only observed with GFP-N, N-128GFP129 and N-130GFP131 (Fig 1B). Thus, GFP insertion in the flexible loop between N residues 106 and 107 or 112 and 113 impeded bioconditensates formation.

We then used FRAP to investigate the dynamics of N inside those NB-like structures. Fluorescence recovery of P-mCherry was similar to what had been published previously [7]. However, there was no fluorescence recovery after photobleaching of GFP-N and N-128GFP129 inside inclusions (Fig 1C). Therefore, the behavior of both proteins within the bioconditensates is different.

## Loops 104–117 and 372–398 on N are required to form bioconditensates in the minimal system

As GFP insertion in the flexible loop L1 extending from residues 104 to 117 (Fig 1A) impeded the ability of N to form bioconditensates in the cellular minimal system (Fig 1B), we decided to investigate the role of this region during the phase separation process.

First, we replaced the residues 104–115 by three consecutive GGGS flexible linkers (*L*) (Fig 1D) (mutant N-103-*3L*-116). Characterization of this mutant reveals that it was expressed (S1C–S1E Fig). However, it was unable to bind P (S1E Fig) and form NB-like structures (Fig 1E). We then tried to identify key features in the segment by constructing mutants N-103-*1L*-108, N-107-*1L*-112 and N-111-*1L*-116 in which only 4 aa-long segments were replaced by a single GGGS motif (Fig 1D). Like N-103-*3L*-116, all those constructions were unable to form cytoplasmic inclusions (Fig 1E), although they were all expressed (S1C and S1D Fig) albeit at a lower level than wild-type as judged by western blot analysis (S1C Fig).

N has a second flexible loop L2 in its C-terminal region (Fig 1A) which binds the C-terminal domain of P when N is associated with RNA as shown for both RABV and VSV [22,23]. We investigated the role of this loop. For this, we constructed the mutants N-373-*5L*-394, N-373-*2L*-382, N-381-*2L*-390 and N-385-*2L*-394 in which stretches of N residues were replaced by 2 or 5 GGGS linkers (*L*) (Fig 1D). Once again, all the mutants failed to form cytoplasmic inclusions (Fig 1E) despite being well-expressed (S1C and S1D Fig) and N-373-*5L*-394 still binding full-length P (S1E Fig). It is worth noting that, unlike full-length P, mutant N-373-*5L*-394 did not co-immunoprecipitate with PΔNTD (residues 53–297), indicating that mutant N-373-*5L*-394 binds the NTD of P and that the loop L2 is indeed involved in binding the CTD of P (S1F Fig).

This indicated that both N loops, 104–117 and 372–398, are required to form bioconditensates in the cellular minimal system.

## N arginines involved in RNA-binding are required for the formation of N-P cellular biocondensates

In infected cells, N is associated with both genomic and antigenomic RNAs. Recombinantly expressed N also binds cellular RNAs to form nucleocapsid-like structures [23,24]. We investigated the role of RNA binding properties of N in its ability to form cytoplasmic inclusions with P. Several arginines of N have been suggested to interact with the phosphate of the RNA backbone [2]. Among them are R149, R168, and R323 (Fig 1F).

Mutations of the three arginines R149, R168, and R323 into alanines abrogated the formation of biocondensates in the presence of P (Fig 1G). We then separately investigated the role of each arginine in this process. Mutation R149A alone abolished the formation of biocondensates while mutants N-R168A and N-R323A kept the ability to phase separate. However, for those two mutants, the concentration of both N and P in the dilute phase (i.e. outside the biocondensates), appeared to be higher than with wild-type N protein (as judged by the intensity of the cytoplasmic fluorescence intensity). Keeping in line with this observation, when both mutations R168A and R323A were introduced, biocondensates were no more observed. The inability of N mutants to form biocondensates was not due to an expression defect: indeed, when expressed alone, all the mutants exhibited a similar homogeneous cellular localization as shown by immunofluorescence (S1G Fig).

Those data indicate that arginines R149, R168 and R323 play a role in biocondensate formation and suggest that N RNA-binding properties are required for the formation of N-P cytoplasmic inclusion bodies.

## Characterization of the phenotype of fluorescent versions of RABV N in infected cells

We also characterized the phenotype of the fluorescent versions of N in infected cells. For this, BSR-T7/5 cells infected with RABV (CVS strain) were transfected 1h post-adsorption with pTit plasmids encoding the different chimeras between N and GFP.

GFP-N was exclusively located inside viral NBs, in which it was homogeneously distributed (Fig 2A). This homogeneous localization inside NBs was in contrast with what is currently observed when N (either GFP-N or N expressed by the virus) localization is analyzed by immunofluorescence using anti-N antibodies which rather reveals ring-like structures (Fig 2A) (see also [6]). Therefore, the apparent peripheral N location observed by immunofluorescence is an artifact.

Unsurprisingly, like GFP-N, N-128GFP129 and N-130GFP131 were localized in the viral NBs (Fig 2B). A more interesting phenotype was observed with N-112GFP113 which partitioned preferentially inside NBs (Fig 2B). This indicates that this N construction is still able to be recruited inside NBs—albeit less efficiently than GFP-N, N-128GFP129, and N-130GFP131, as judged by the remaining fluorescence intensity in the cytoplasm—despite its inability to induce biocondensates formation in the minimal system (Fig 1B). Finally, N-106GFP107 remained in the cytosol and did not concentrate inside NBs (Fig 2B). However, cells expressing N-106GFP107 were less likely to develop a productive infection. Indeed, for similar percentages of infected and transfected cells, the proportion of simultaneously infected and transfected cells was much lower when cells were transfected with pTit-N-106GFP107 than by pTit-GFP-N (Fig 2C and 2D). This suggests that somehow the N-106GFP107 protein exerts a dominant-negative effect on RABV infection.

We then fused the GFP to the amino-terminus of N-373-*5L*-394 mutant that is unable to form biocondensates in presence of P. Surprisingly, this chimera also behaves as GFP-N and was homogenously distributed inside NBs (Fig 2B). Thus, the ability of some N mutants to

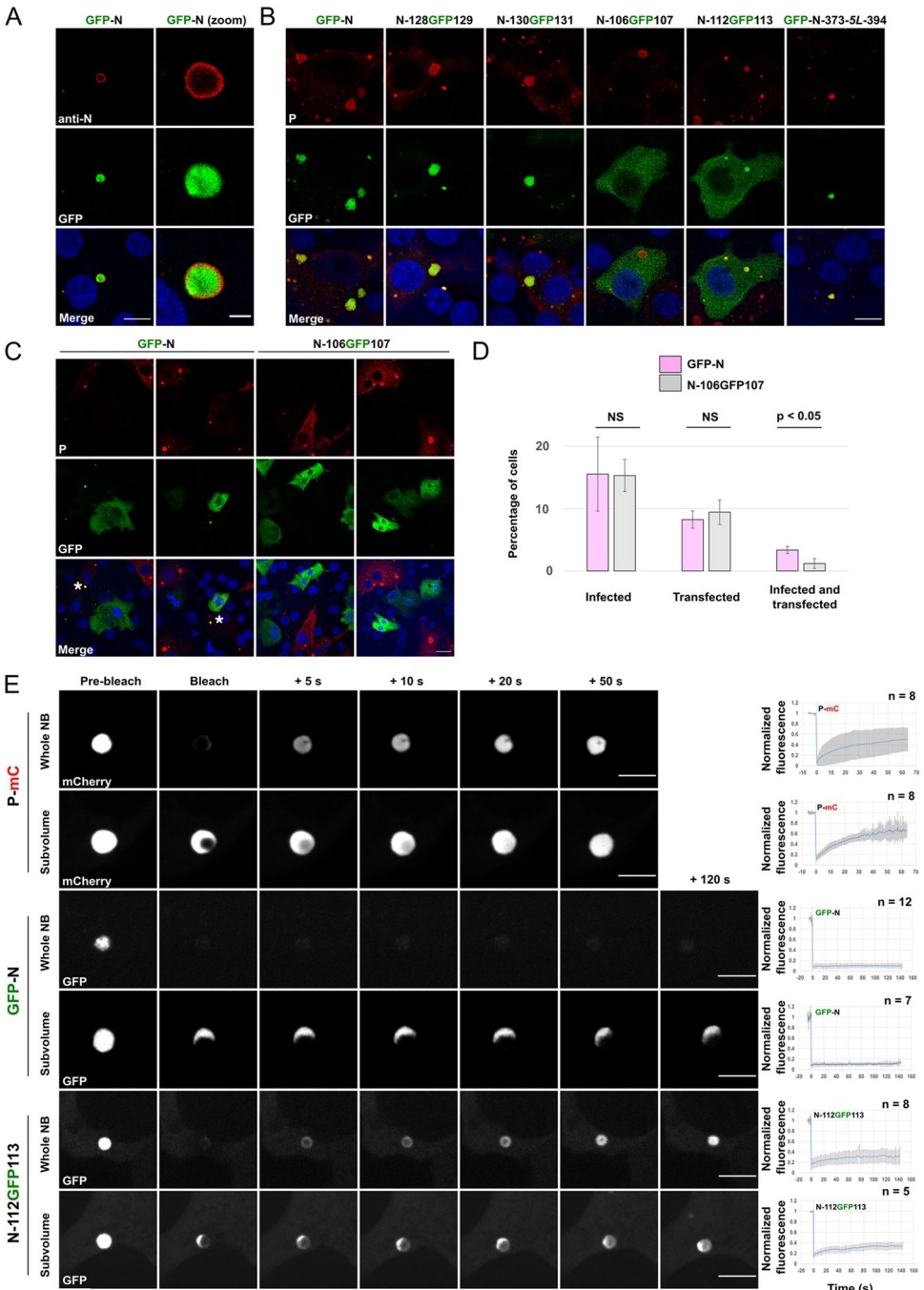

**Fig 2. Characterization of the behavior of chimeras between GFP and N in RABV infected cells. A-C)** BSR-T7/5 cells were infected at an MOI of 0.5 by RABV (CVS strain) and transfected 1h post adsorption with pTit plasmids encoding the indicated N constructs. Cells were fixed 14 h post transfection and the localization of viral proteins and ectopically expressed N mutants was assessed by fluorescence confocal microscopy. **A)** N protein (either virally encoded or in the GFP-N chimera) was revealed with a mouse monoclonal anti-N antibody followed by incubation with Alexa-488 goat anti-mouse IgG. Scale bars: 10 μm and 2μm for the magnified image. **B)** Localization of viral P protein (revealed with a rabbit polyclonal anti-P antibody followed by incubation with Alexa-568 donkey anti-rabbit IgG) and GFP-N chimeras in RABV infected cells. Scale bar: 10 μm. **C)** Effect of GFP-N and N-106GFP107 expression on RABV cell infection. P was revealed with a rabbit polyclonal anti-P antibody followed by incubation with Alexa-568 donkey anti-rabbit IgG. Asterisks indicate cells that are infected and express GFP-N. Note that GFP-N localization is diffuse in the cells that are not infected. Infected cells expressing N-106GFP107 are rare and absent in the displayed field. Scale bar: 10 μm. **D)** Percentages of infected, transfected, as well as both infected and transfected cells (for GFP-N

and N-106GFP107 chimeras) were assessed from experiments such as those presented in Fig 2C. Values are the average of 3 independent experiments ± SD. The total number of cells analyzed was 1555 (353 + 407 + 795) for GFP-N and 1804 (241 + 615 + 948) for N-106GFP107. p-values were calculated using two-tailed Student's t test. **E)** NBs containing P-mCherry, GFP-N or N-112GFP113 constructs were photobleached and the recovery of fluorescence was imaged with a spinning disk microscope. Whole NBs as well as NBs' subvolumes were photobleached. Scale bars: 5 μm. For the plots on the right, FRAP data were corrected for background, normalized to the minimum and maximum intensity. The mean is shown with error bars representing the SD.

induce the formation of biocondensates seems to be distinct from their propensity to accumulate inside viral NBs.

Finally, we performed FRAP experiments on those fluorescent versions of N, expressed after transfection of infected cells, to investigate their dynamics inside NBs. We photobleached defined regions inside a NB as well as entire NBs. Here again, unlike for P-mCherry, there was no fluorescent recovery after photobleaching of GFP-N. However, a partial fluorescent recovery was observed for N-112GFP113 (Fig 2E).

## Quantification of P and N concentrations in the dilute and dense phases

As mentioned above, N-R168A and N-R323A mutants, even if they were still able to phase-separate in the presence of P, seemed to behave differently from the wild-type N. To further document this, we attempted to determine the concentrations of N and P in the dilute and dense phases as described in [25]. We used fluorescent versions of the proteins and assumed that the local concentration of a protein is proportional to its local fluorescence intensity per pixel.

First, we took advantage of the minimal system and co-expressed GFP-N and P-mCherry proteins in BSR-T7/5 cells. We measured the areal fluorescence for N and P in the dilute phase ($f_N$ dilute and $f_P$ dilute) at several locations in the cytosol for 10 different cells (Fig 3A). We observed a slight intracellular variation which may reveal some cytoplasmic heterogeneity. The intercellular variation was more important (see cells 5 and 7 in Fig 3A). Finally, for both proteins, there was no correlation between their areal fluorescence measured in the dense and in the dilute phases (Fig 3B).

We then compared the behavior of chimeras GFP-N, N-128GFP129, and N-130GFP131 when co-expressed with P-mCherry. For this, we plotted the areal fluorescence of GFP-N *vs* the areal fluorescence of P-mCherry in the dilute (Fig 3C) and in the dense (Fig 3D) phases. For each chimera, we observed a positive correlation between the GFP-N and P-mCherry areal fluorescences. However, there were no significant differences between the chimeras (Fig 3C–3E).

We also compared the phenotype of mutants GFP-N-R168A and GFP-N-R323A to that of GFP-N (Fig 3F and 3H). For both mutants, we observed a significant increase of P and N concentrations in the dilute phase (Fig 3F and 3H) consistent with the observations in Fig 1G. Nevertheless, for both mutants, the plots of $f_N$ dense *vs* $f_P$ dense were very similar to that of GFP-N (Fig 3G and 3H). Overall, this suggests that mutant proteins N-R168A and N-R323A require higher concentrations of P and N to phase-separate (Fig 3F and 3H) consistent with the observations in Fig 1G.

Second, we tried to perform similar quantifications in the infectious context. For this, we infected BSR-T7/5 cells with a recombinant virus in which the P gene has been replaced by that encoding P-mCherry (rCVS-P-mCherry described in [26]) before transfection with a plasmid expressing GFP-N. When we measured the areal fluorescence for N and P in the dilute phase at several locations in the cytosol of infected cells (Fig 3I), we observed greater intercellular variations than in the minimal system. This might be related to the presence of viral N

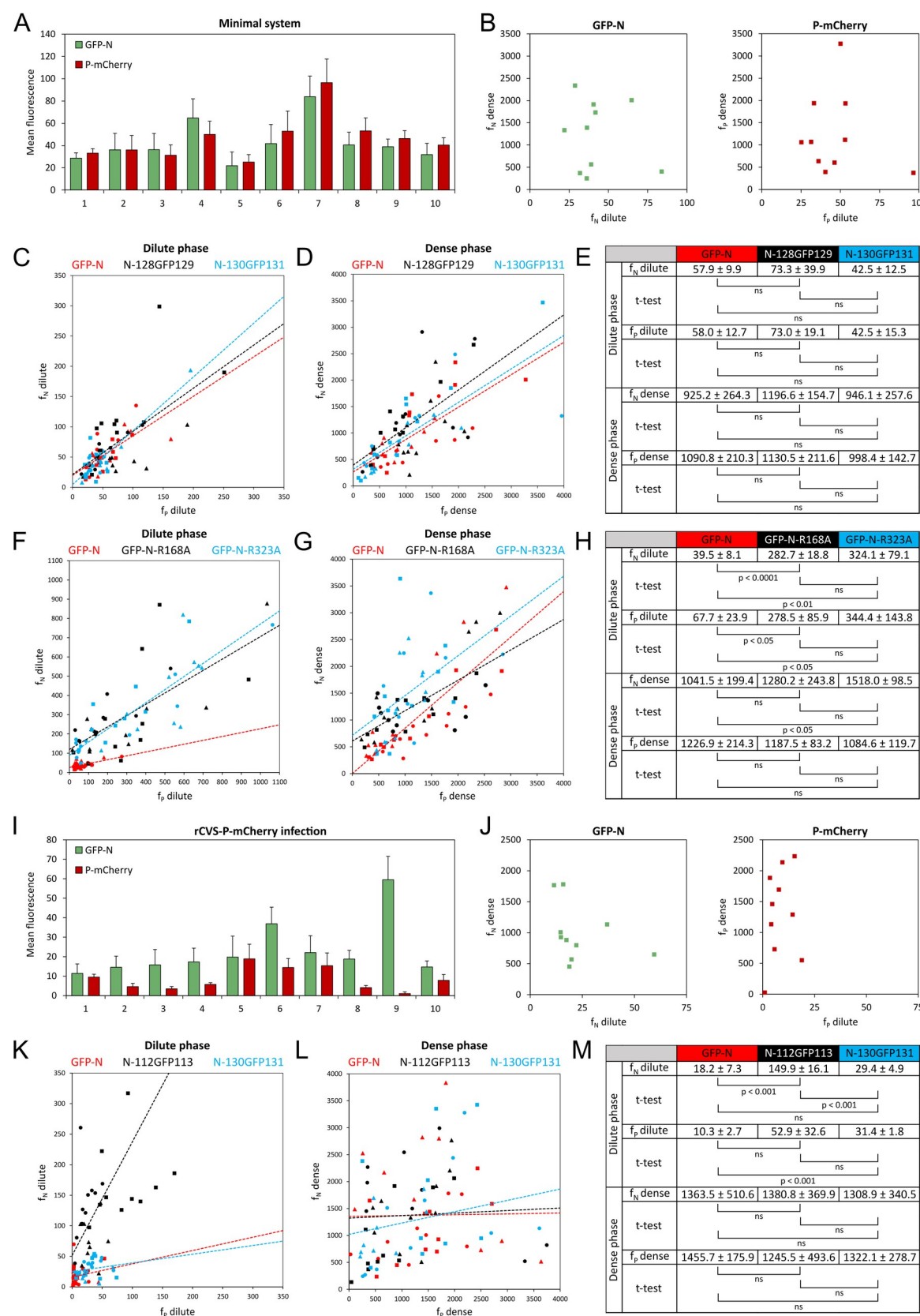

**Fig 3. Quantification of N and P concentrations in the dilute and dense phases. A-H)** BSR-T7/5 cells were co-transfected with pTit plasmids encoding P-mCherry protein and the indicated fluorescent N constructs (GFP-N, N-128GFP129, N-130GFP131, GFP-N-R168A and GFP-N-R323A). Cells were fixed 24 h post-transfection and the concentrations of ectopically expressed P and N (wild-type and mutants) were determined by measuring fluorescence intensity from confocal images. **A)** Histogram showing the mean (± SD) areal fluorescence (calculated from 5 different locations for each cell) of both GFP-N and P-mCherry in the dilute phase for 10 random cells. **B)** Plot of areal fluorescence in dense *vs* dilute phase of GFP-N (left) or P-mCherry (right) for 10 random cells. **C-D)** Plot of areal fluorescence of N constructs (GFP-N, N-128GFP129 and N-130GFP131) *vs* areal fluorescence of P-mCherry in dilute **(C)** and dense **(D)** phases. Data points represent 3 independent experiments (10 cells were quantified per independent experiment). Data points from the same experiment are represented by the same symbol (squares, circles and triangles). The dashed lines represent the trend. **E)** Table giving the mean (± SD) of the average areal fluorescence calculated for the indicated GFP-N constructs and P-mCherry in both dilute and dense phases in each 3 independent experiment. P-values were calculated using a Student t-test. **F-G)** Plot of areal fluorescence of N constructs (GFP-N, GFP-N-R168A and GFP-N-R323A) *vs* areal fluorescence of P-mCherry in dilute **(F)** and dense **(G)** phases. Data points represent 3 independent experiments (10 cells were quantified per independent experiment). Data points from the same experiment are represented by the same symbol (squares, circles and triangles). The dashed lines represent the trend. **H)** Table giving the mean (± SD) of the average areal fluorescence calculated for the indicated GFP-N constructs and P-mCherry in both dilute and dense phases in each 3 independent experiment. P-values were calculated using a Student t-test. **I-M)** BSR-T7/5 cells were infected at an MOI of 0.5 by rCVS-P-mCherry and transfected 1h post adsorption with pTit plasmids encoding the indicated fluorescent N constructions (GFP-N, N-112GFP113 and N-130GFP131). Cells were fixed 14 h post-transfection and the concentrations of fluorescent versions of P and N were determined by measuring fluorescence intensity from confocal images. **I)** Histogram showing the mean (± SD) areal fluorescence (calculated from 5 different locations for each cell) of both GFP-N and P-mCherry in the dilute phase for 10 random cells. **J)** Plot of areal fluorescence in dense *vs* dilute phase of GFP-N (left) or P-mCherry (right) for 10 random cells. **K-L)** Plot of areal fluorescence of N constructs (GFP-N, N-112GFP113 and N-130GFP131) *vs* areal fluorescence of P-mCherry in dilute **(K)** and dense **(L)** phases. Data points represent 3 independent experiments (10 cells were quantified per independent experiment). Data points from the same experiment are represented by the same symbol (squares, circles and triangles). The dashed lines represent the trend. **M)** Table giving the mean (± SD) of the average areal fluorescence calculated for the indicated GFP-N constructs and P-mCherry in both dilute and dense phases in each 3 independent experiments. P-values were calculated using a Student t-test.

protein (the concentration of which is unknown and probably varies from cell to cell). Once again, for both proteins there was no correlation between their areal fluorescence measured in the dense and in the dilute phases (Fig 3J). Nevertheless, chimera N-112GFP113 and to a lesser extent N-130GFP131 exhibited a distinct phenotype from that of chimera GFP-N. In the case of chimera N-112GFP113, there was a significant increase of N concentration in the dilute phase (Fig 3K and 3M) consistent with the observations in Fig 2B. Conversely, in the case of chimera N-130GFP131, there was a slight but significant increase of P concentration in the dilute phase (Fig 3K and 3M). These concentration differences in the dilute phase were not associated with significant concentration changes in the dense phase (Fig 3L and 3M).

## RABV P alone phase separates *in vitro* in crowded environments

As IDD-containing proteins involved in cellular biocondensates formation can undergo LLPS in acellular system [8,12,27–30], we investigated if this could be the case for RABV P.

For this, recombinant RABV P (PV strain), having a C-terminal StrepTagII and in which residue Cys261 has been replaced by a serine (so that the resulting protein has a single cysteine residue in position 297) (Fig 4A), was expressed in *E. coli* and purified (Fig 4B). As expected [19,20], purified P is a dimer as shown by SEC-MALS analysis (S2A Fig). The A260/A280 ratio was 0.59 indicating that the protein was nucleic acids free (S2B Fig). In presence of 5% PEG 8000, at 10μM concentration, purified recombinant P forms spherical structures, having approximatively 1 to 5μm in diameter, that were visible using differential interference contrast (DIC) microscopy (Fig 4C). We also expressed and purified full-length P using a recombinant baculovirus in Hi-5 lepidoptera cells (Fig 4B). At 5μM concentration, it also forms spherical droplets in presence of 5% PEG 8000 (Fig 4C). This indicated that the expression system of P, whether prokaryotic or eukaryotic, had no impact on the ability of purified P to phase separate in crowded environments.

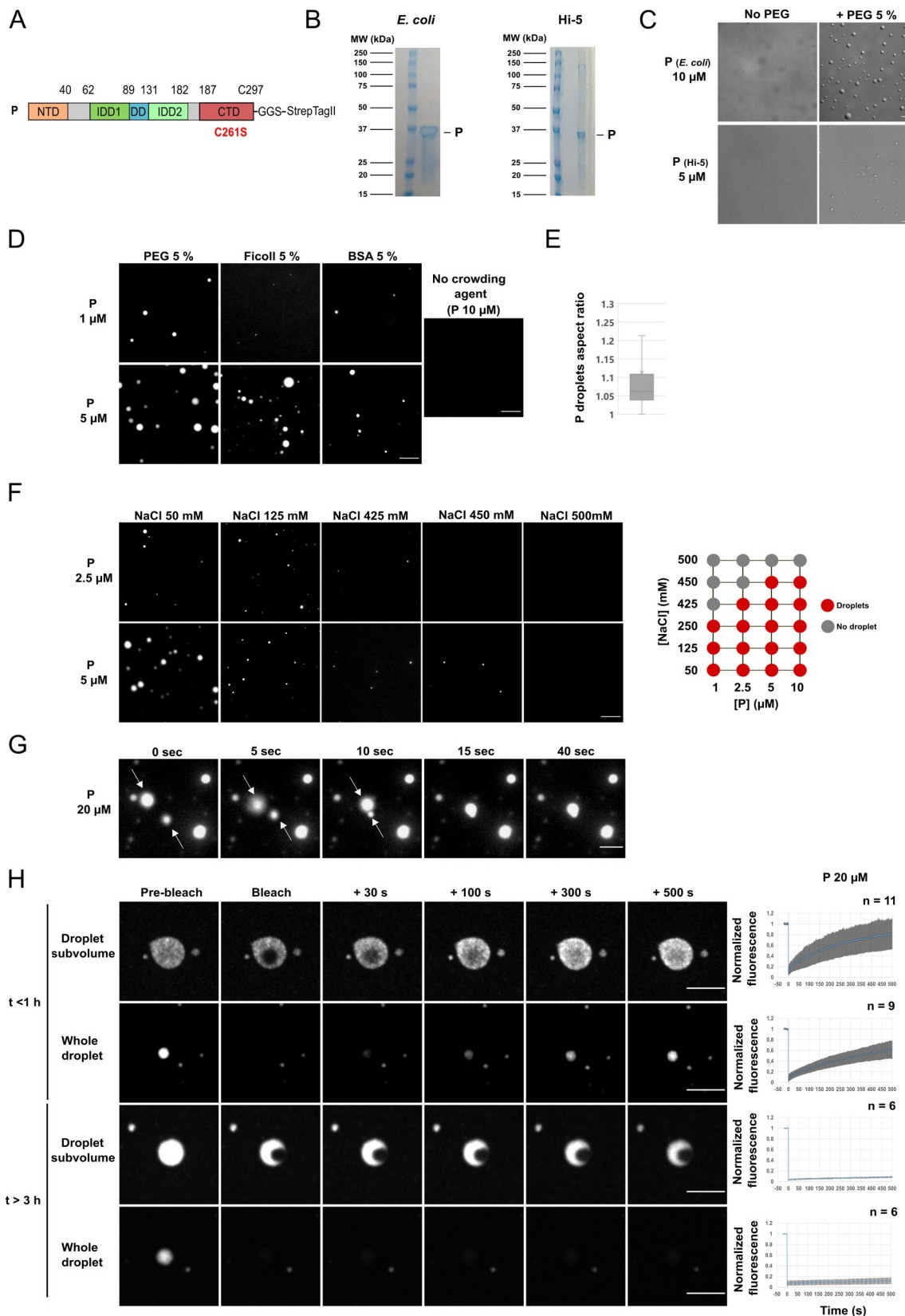

**Fig 4. RABV P alone phase separates in crowded environments. A)** Schematic representation of the organization of RABV P construct used. P contains an N-terminal domain (NTD) that associates with N in the N0-P complex and with the viral polymerase, two intrinsically disordered domains (IDD1 and IDD2), a dimerization domain (DD) and a C-terminal domain (CTD) that binds N associated with RNA. The cysteine at position 261 was replaced by a serine so that the resulting protein has only one cysteine residue in position 297. A StrepTagII was used to purify the protein. **B)** SDS PAGE analysis of purified P proteins. **C)** P-Strep protein expressed in *E. coli* at 10 μM concentration (320 μg/ml) (resp. full-length P protein expressed in Hi-5 cells at 5 μM) was incubated in absence or presence of 5% PEG 8000 in 125 mM NaCl, 20 mM Tris-HCl pH7.5. Droplets were imaged by differential interference contrast (DIC). Scale bar: 10 μm. **D)** P-Strep droplets were observed by fluorescence microscopy at different P concentrations in absence or presence of different molecular crowders in 50 mM NaCl, 20 mM Tris-HCl pH7.5. The mix contained 25 nM of P protein covalently labelled with Cy3 maleimide probe. Scale bars: 10 μm. **E)** Box plot representation of P-Strep droplets aspect ratios (formed at 5 μM with 5% PEG8000 as in D). Sample size: n = 936. The cross on the boxplot indicates the mean. **F)** Left panel: P-Strep droplets were observed by fluorescence microscopy at different concentrations in presence of 5% PEG 8000 and increasing concentrations of NaCl in 20 mM Tris-HCl pH 7.5 buffer. The mix contained 25 nM of P protein covalently labelled with Cy3 maleimide probe. Scale bars: 10 μm. Right panel: phase diagram at various P-Strep concentrations and with increasing concentrations of NaCl in presence of 5% PEG 8000. **G)** Fusion between P-Strep droplets ([P] = 20 μM in the presence of 3% PEG 8000, 125 mM NaCl, 20 mM Tris-HCl pH 7.5) imaged by time-lapse video-microscopy. Images were extracted from S2 Movie at the indicated times. Scale bar: 2 μm. **H)** Droplets formed by P-Strep at the concentration of 20 μM (50 nM of fluorescent P protein) in the presence of 5% PEG 8000 in 125 mM NaCl, 20 mM Tris-HCl pH 7.5 buffer were photobleached at different time-points post-mixing (less than 1 h or more than 3 h). Whole droplets as well as droplets subvolumes were photobleached. Images were acquired on a spinning disk microscope. For the plots on the right, FRAP data were background-corrected and normalized to the minimum and maximum intensity. The mean is shown with error bars representing the SD. Scale bars: 5 μm.

Phase separation was also analyzed using RABV P covalently-labeled on Cys297 with Cy3 (Fig 4D and S1 Movie). Regardless of the crowding agent used (5% Ficoll-400, 5% bovine serum albumin -BSA-, 5% PEG 8000), in the presence of 50 mM NaCl, fluorescent spherical droplets (Fig 4E) were observed at total P concentration as low as 1 μM (Fig 4D). In the absence of crowding agent, no phase separation was observed in 50 mM NaCl at a P concentration of 10 μM (Fig 4D). Further investigations were thus conducted in presence of 5% PEG 8000.

The phase transition appeared to be salt-dependent. In the presence of 425 mM NaCl, rare and smaller fluorescent droplets were still observed at a P concentration of 2.5 μM. In the presence of 450 mM, droplets were only observed from 5 μM. Finally, no droplets were observed in the presence of 500 mM NaCl (Fig 4F).

The spherical aspect of the droplets (Fig 4D and 4E) suggested that they were formed by LLPS. This was confirmed by visualizing several fusion events between two droplets followed by relaxation into a bigger spherical structure (Fig 4G and S2 and S3 Movies). Finally, we performed FRAP experiments. We photobleached defined regions inside a droplet as well as whole droplets. In both cases, on droplets less than one hour old, fluorescence recovery was almost complete in about 10 minutes with a half-time in the range of ~100 s (Fig 4H). However, on droplets older than 3h, fluorescence recovery was nearly abrogated, suggesting that the initially liquid-like droplets had transitioned to a gel-like state during the experiment.

Taken together, these data demonstrate that P alone, at low concentration, induces LLPS *in vitro* in crowded environments. This is different from what is observed in the cellular minimal system, in which both N and P are required for the formation of liquid inclusions.

## P CTD but not P NTD is required for P phase separation *in vitro*

In the cellular minimal system, P CTD is essential for the formation of liquid inclusions in the presence of N and deletion of P region [195–297] abolishes inclusion formation. On the other hand, the amino-terminal part of P is not required as deletions corresponding to domains [1–52] or [62–86] do not impede inclusions formation [7]. Therefore, we analyzed the requirement of P CTD and P NTD for P phase separation *in vitro*.

P deletion mutants (Fig 5A) were expressed in *E. coli* (Fig 5B). The production yield of PΔCTD was like that of full-length P whereas the yield of PΔNTD was about 5 fold lower.

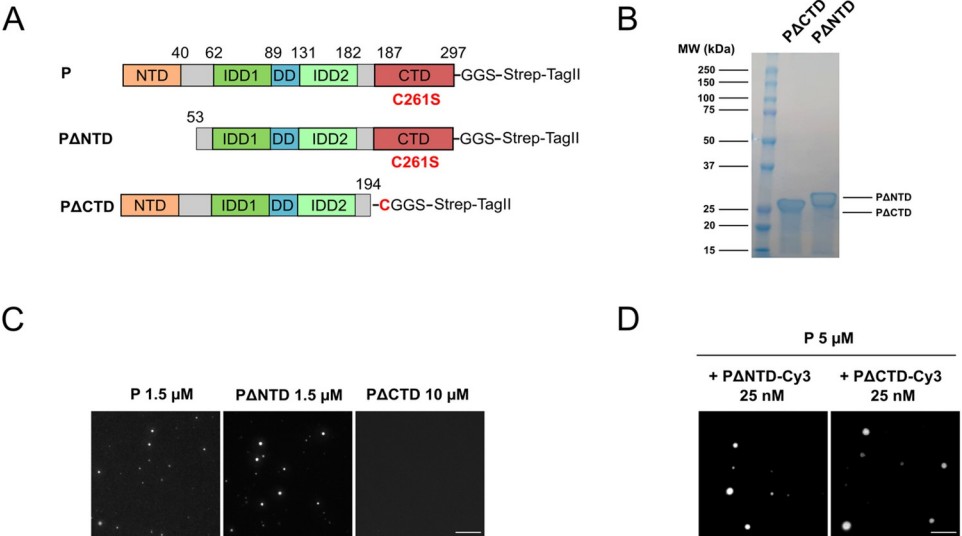

**Fig 5. Phase separation of RABV P deletion mutants. A)** Schematic representation of the domain organization of RABV P deletion mutants used. The PΔNTD construct is deleted from amino acids 1–52. The PΔCTD construct is deleted from amino acids 195–296. **B)** SDS PAGE analysis of purified PΔCTD and PΔNTD. **C)** P deletion mutants droplets were observed by fluorescence microscopy at different concentrations in presence of 5% PEG 8000 in 125 mM NaCl, 20 mM Tris-HCl pH 7.5 buffer. The mix contained 25 nM of fluorescent P deletion mutant. Scale bar: 10 μm. **D)** Non-fluorescent P at 5 μM was incubated with either fluorescent PΔNTD or fluorescent PΔCTD constructs at 25 nM in presence of 5% PEG 8000 in 50 mM NaCl, 20 mM Tris-HCl pH 7.5 buffer. Presence of droplets was assessed by fluorescence microscopy. Scale bar: 10 μm.

However, both deletion mutants were soluble and formed dimers in solution as determined by SEC-MALS analysis (S2C and S2D Fig). In presence of 5% PEG 8000, PΔNTD was shown to phase separate as wild-type P at micromolar concentrations (Fig 5C). No phase separation was observed with PΔCTD at a concentration of 10 μM (Fig 5C). Thus, the same domains are required for P to phase separate *in vitro* and in the cellular minimal system in presence of N suggesting that both phenomena involve the same physicochemical principles and molecular bases.

We also mixed fluorescent PΔNTD or PΔCTD (25 nM) with 5 μM of non-labelled P in presence of 5% PEG 8000 (Fig 5D). Both deletion mutants appeared to be concentrated in the P droplets. Particularly, PΔCTD, although unable to phase separate *per se*, is nevertheless recruited inside the droplets.

## RABV P droplets concentrate RNA, exclude BSA and associate with liposomes

As several protein condensates have been shown to concentrate RNA molecules, we incubated polyA RNA covalently labelled on their 5' end with Cy-5 fluorescent dye. PolyA RNA of 10 and 40 nucleotides (100 nM) concentrated in P droplets in the presence of 5% PEG 8000 while they were unable to phase separate without P (Fig 6A). On the other hand, fluorescent BSA was excluded from the droplets (Fig 6A).

We analyzed the diffusion of the RNA molecules within the droplets by FRAP (Fig 6B). Here again, we photobleached defined regions within a droplet as well as entire droplets. In both cases, for droplets less than one hour old, fluorescence recovery was fast and complete ($t_{1/2}$ within the second range). However, after 4 hours of incubation, recovery was slowed for 10-nt RNA and almost abrogated for 40-nt RNA.

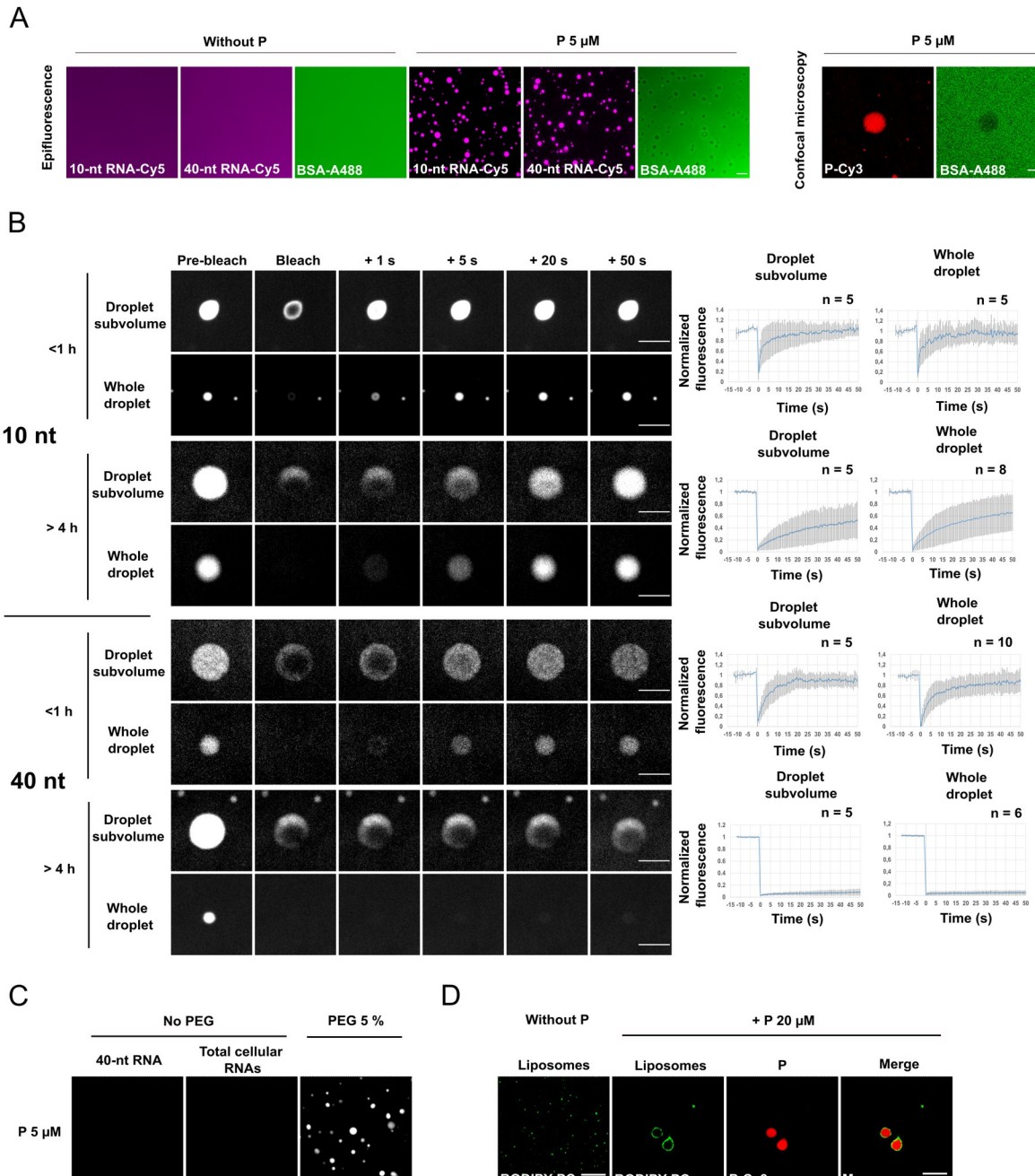

**Fig 6. Biologically relevant properties of RABV P droplets. A)** Segregation of RNA-Cy5 (10- or 40-A nucleotides) at the concentration of 100 nM into non-fluorescent P droplets (P at 5 μM in presence of 5% PEG 8000 in 125 mM NaCl, 20 mM Tris-HCl pH 7.5) and exclusion of BSA-FITC at the concentration of 250 μg/ml (3.8 μM) from P droplets. Fluorescent RNA and BSA were added when LLPS was induced. Analysis was performed by fluorescence microscopy. Exclusion of BSA-FITC from P droplets was validated by confocal microscopy (right panel). Scale bars: 10 μm. **B)** Droplets formed by non-fluorescent P at the concentration of 20 μM in the presence of 5% PEG 8000 and 100 nM Cy5-RNAs in 125 mM NaCl, 20 mM Tris-HCl pH 7.5 were photobleached at different time-points post-mixing (less than 1 h or more than 4 h). Whole droplets as well as droplets subvolumes were photobleached. Images were acquired on a spinning disk confocal microscope. Scale bars: 5 μm. For the plots on the right, FRAP data were corrected for background, normalized to the minimum and maximum intensity. The mean is shown with error bars representing the SD. **C)** P at 5 μM (25 nM of fluorescent P protein) in 50 mM NaCl, 20 mM Tris-HCl pH 7.5 was incubated with either 5% PEG 8000, or 80 ng/μL (6 μM) of synthetic 40-A nucleotides RNA, or 80 ng/μL of total RNAs extracted from BSR-T7/5 cells. Presence of droplets was assessed by fluorescence microscopy. Scale bar: 10 μm. **D)** Fluorescent liposomes (PE/PC/PS = 2/2/1 w/w, 100 μg/ml *i.e.* ~130 μM of lipids containing 0.25% of BODIPY 500/510 C12-HPC) were incubated or not with P at 20 μM concentration (50 nM of fluorescent P protein) in 5% PEG 8000, 125 mM NaCl, 20 mM Tris-HCl pH 7.5 before analysis by confocal microscopy. Scale bars: 10 μm.

RNA, at concentrations of a few tens of ng/μl, has been shown to induce LLPS of some proteins even in the absence of crowding agents [29,31]. However, at the concentration of 80 ng/μl, neither synthetic 40-nt RNA nor total cellular RNAs induce P phase separation (Fig 6C).

At late stages of infection, membranes, most probably ER-derived, associate with NBs [6,18]. For this reason, we investigated the interaction between P droplets and liposomes composed of phosphatidylcholine, phosphatidylethanolamine and phosphatidylserine (w/w/w: 2/2/1). We observed that the liposomes spontaneously associated with and surrounded the P droplets (Fig 6D).

## N0-P but not purified nucleocapsids phase separates *in vitro*

N exists in two main states in the cell: (i) the N0-P complex, in which, upon binding the amino-terminal region of P, N cannot self-associate nor bind RNA, and (ii) the helical nucleocapsid in which N is tightly associated with genomic or antigenomic RNAs [2,5,23]. The N and P were co-expressed in insect cells using recombinant baculoviruses and N0-P complex was purified to homogeneity thanks to the amino-terminal StrepTag of N and removal of RNPs by gel filtration (Fig 7A and 7B). The nucleocapsids from RABV PV strain were purified from infected BSR cells.

The N0-P complex phase separates under similar conditions as P alone (Fig 7C), although the condensates formed appear to be less spherical (compare Figs 4E and 7D). Addition of small amounts of fluorescently-tagged RABV P revealed that P concentrates within the N0-P dense phase (Fig 7E) and, similarly, fluorescently-labelled N0-P concentrates in P droplets (Fig 7E). Finally, FRAP experiments performed on N0-P droplets revealed that fluorescence recovery was a slow process with a $t_{1/2}$ greater than 5 minutes (Fig 7F).

Conversely, purified nucleocapsids alone (Fig 7G) did not form liquid droplets even in the presence of PEG 8000 (Fig 7H). They formed non-spherical heterogeneous structures. However, addition of increasing amounts of P protein (containing 25 nM of fluorescent P) revealed that P bound non-homogeneously to the nucleocapsids which, however, retained their amorphous structures. When P was in excess, P droplets were formed that seem to exclude the nucleocapsids (Fig 7H).

We also mixed RNA with the N0-P complex either ten minutes before or just when adding the crowding agent, or ten minutes after the onset of phase separation. We observed that RNA associated with N0-P droplets (Fig 7I). However, the spherical shape of the droplets was rapidly lost (Fig 7J and 7K). This suggests that association of N with RNA modifies its properties, which leads to a modification of the properties of N0-P droplets.

## Discussion

The discovery that MNV factories have liquid properties and are formed by LLPS raises several questions about the organization and properties of those viro-induced compartments, the transcription and replication processes in such an environment, and the release of RNPs from these cytoplasmic biocondensates [7,13,15,17,32]. To decipher the molecular bases underlying such processes, it is necessary to design versatile and easy-to-handle minimal systems that recapitulate the characteristics of these viral biocondensates. Here, we describe and compare two minimal systems, a cellular and an acellular one, that mimic NB properties.

### Characterization of N behavior in the cellular minimal system

P and N, when co-expressed by transfection are able to form biocondensates [7]. This minimal system was previously used to investigate the dynamics of P inside these NBs-like structures as well as P domains required for LLPS. However, nothing was known about the behavior of N.

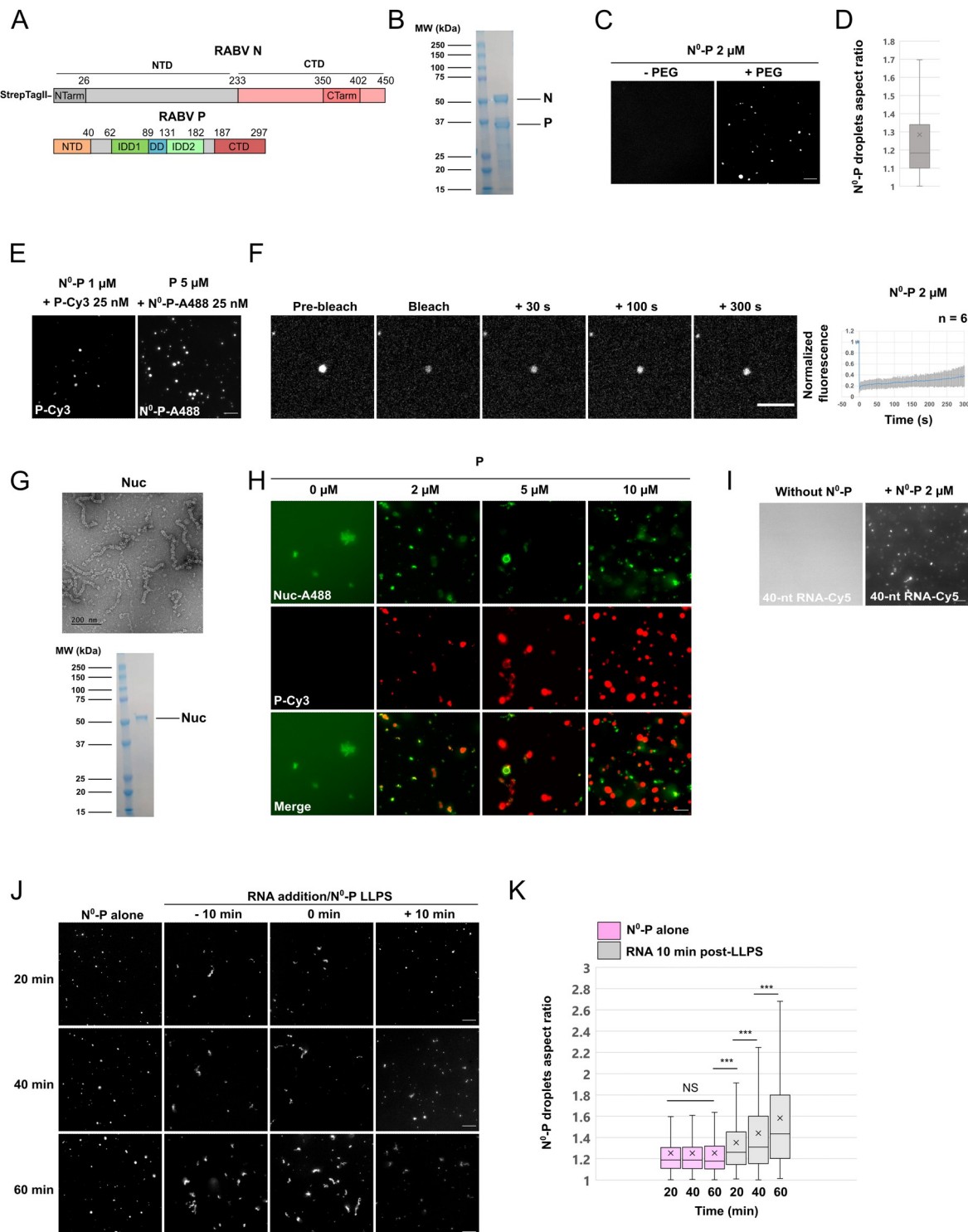

**Fig 7. Characterization of the behavior of RABV N in acellular minimal systems. A)** Schematic representation of the organization of RABV N and P constructs used. A StrepTagII was amino-terminally fused to the N protein to purify the N0-P complex from insect cells. Domains delimitations are indicated for both proteins. **B)** SDS PAGE analysis of N0-P complex. **C)** Purified N0-P complex labelled with NHS-Ester-Atto-488, was incubated at 2 μM concentration in 50 mM NaCl, 20 mM Tris-HCl pH 7.5 in absence or presence of 5% PEG 8000. Presence of N°-P droplets was assessed by fluorescence microscopy. Scale bar: 10 μm. **D)** Box plot representation of N°-P droplets aspect ratios (formed at 2 μM with 5% PEG 8000 as in C). Sample size: n = 1707. The cross on the boxplot indicates the mean. **E)** Fluorescent P (resp. fluorescent N0-P) at 25 nM was incubated with 1 μM N0-P (resp 5 μM P) in 50 mM NaCl, 20 mM Tris-HCl pH 7.5

and 5% PEG 8000. Presence of droplets was assessed by fluorescence microscopy. Scale bar: 10 μm. **F)** Whole droplets formed by N0-P complex at the concentration of 2 μM (25 nM of fluorescent complex) in the presence of 5% PEG 8000 in 50 mM NaCl and 20 mM Tris-HCl pH 7.5 were photobleached. Images were acquired on a spinning disk microscope. Scale bars: 5 μm. For the right plot, FRAP data were corrected for background, normalized to the minimum and maximum intensity. The mean is shown with error bars representing the SD. **G)** Top: Electron microscopy image of negatively stained viral nucleocapsids (Nuc), purified from RABV-infected BSR cells. Bottom: SDS PAGE analysis of purified nucleocapsids. **H)** Purified nucleocapsids (500 nM of N protein, 10% fluorescently-labelled), labelled with NHS-Ester-Atto-488 were incubated in absence or presence of increasing concentrations of P (125 mM NaCl, 20 mM Tris-HCl pH 7.5, 5% PEG 8000, 25 nM of fluorescent P). Nucleocapsids and P droplets were observed by fluorescence microscopy. Scale bar: 10 μm. **I)** Segregation of RNA-Cy5 (40-A nucleotides) at the concentration of 100 nM into non-fluorescent N0-P droplets (P at 2 μM in presence of 5% PEG 8000 in 125 mM NaCl, 20 mM Tris-HCl pH 7.5). Analysis was performed by fluorescence microscopy. Scale bars: 10 μm. **J)** N°-P droplets (2 μM concentration among which 25 nM of fluorescent complex, 5% PEG 8000, NaCl 50 mM, 20 mM Tris-HCl pH 7.5) in absence of RNA or incubated in presence of 40-A RNA added either 10 min before PEG-induced phase separation, or at the time of PEG 8000 addition, or 10 min after PEG-induced phase separation were observed by fluorescence microscopy. Images were acquired 20, 40, and 60 min after LLPS induction. Scale bars: 10 μm. **K)** Boxplot analysis of N0-P droplets aspect ratios in absence or presence of RNA added 10 min after LLPS. The time indicated on the left corresponds to the time elapsed since LLPS induction. From left to right, the sample size in each category is n = 536, n = 1065, n = 1087, n = 694, n = 962, n = 882. The cross on each boxplot indicates the mean. P-values were calculated using Kolmogorov-Smirnov test. ** $p < 2.10^{-4}$, *** $p < 10^{-6}$.

In this work, we constructed fluorescent N proteins that are able to phase separate when co-expressed with P and analyzed their dynamics by FRAP. The behavior of N is quite different from that of P since there is no fluorescence recovery after photobleaching of GFP-N and N-128GFP129 inside inclusions. It is thus likely that in such N-P inclusions, N is stably associated with cellular RNAs and forms N-RNA rings [2] and RNP-like structures [23,24] which are poorly mobile inside the biocondensates.

When GFP is inserted in the loop L1 extending from residues 104 to 117 of N, the resulting fluorescent N is unable to form biocondensates with P. This suggested that loop L1 was involved in the phase separation. Indeed, replacements of L1 segments with GGGS motifs abolish N ability to phase separate with P. Similarly, loop L2 extending from residues 373 to 394 is required for phase separation. Both loops are absent in the N crystal structure, suggesting that they are flexible and might be disordered. The implication of such disordered regions in LLPS is well documented in several biological systems [8,12,27–29]. Nevertheless, since L2 also constitutes the P binding site when N is associated with RNA [22], the precise property of the L2 loop involved in LLPS remains to be determined. Interestingly, mutations of arginines that contact the phosphodiester backbone of the RNA molecule (and thus are directly involved in N-RNA binding) also abolish LLPS. This suggests that N-RNA association is important for biocondensates formation in the cellular minimal system.

When expressed in previously infected cells, GFP-N, N-128GFP129, N-130GFP131 are localized in viral NBs. Their homogeneous distribution in the viral factory is in contrast to their seemingly annular distribution revealed by an anti-N antibody. This again shows that extreme caution should be the rule when interpreting the organization of liquid biocondensates from imaging data obtained using antibodies in fixed cells.

Unexpectedly, an NB localization is also observed for N-112GFP113 and GFP-N-373-*5L*-394 which are unable to induce biocondensates formation in the cellular minimal system. This establishes a functional distinction between N ability to act as a core protein and to be recruited inside preformed biocondensates. In other words, some mutations transform N in a simple client protein of NBs. Finally, the N-106GFP107 mutant has a dominant-negative effect on NB formation and prevents the development of the infectious cycle in the cells in which it is expressed.

As in the minimal system, in infected cells, when NBs containing GFP-N are photobleached, there is no fluorescence recovery inside the inclusions. However, a partial but significant fluorescence recovery is observed when NBs containing N-112GFP113 are photobleached. This indicates that N-112GFP113 can shuttle between the cytoplasm and NBs.

Keeping in line with this idea, the concentration of N-112GFP113 in the dilute phase is higher than that of GFP-N.

## Towards a thermodynamic characterization of intracellular N-P phase separation?

Using an approach similar to that of Riback and colleagues [25], we quantified the concentration of P and N in the dilute and dense phases. This allowed us, for example, to demonstrate that the behaviors of GFP-N-R168A and GFP-N-R323A are distinct from those of GFP-N in the cellular minimal system and that they phase separate at higher P and N concentrations.

Nevertheless, the correlation between P and N concentrations in the dilute phases (simultaneously high or simultaneously low) is unexpected. This suggests that in many cases (when the dilute phase concentrations are high), we are far from equilibrium or that the state of the cell has a considerable influence on the phenomenon. In any case, the extraction of thermodynamic parameters from these data seems very difficult if not impossible.

## Purified P and N0-P form liquid biocondensates in crowded environments

In most *in vitro* systems mimicking inclusions formed upon infection with other *Mononegavirales* [14,16], LLPS was induced using nucleoprotein-phosphoprotein complexes [14,16]. Here, we show that RABV P alone can phase separate and form biocondensates in several crowded environments (5% PEG 8000, 5% Ficoll 400, 5% BSA), similar to the P proteins of human metapneumovirus and barley yellow striate mosaic virus [30,33]. P droplets formation by LLPS is consistent with their spherical structure, their ability to fuse together, and the FRAP data acquired at an early time after their formation. P critical concentration to trigger LLPS is below 1 μM (i.e. below 30 μg/ml) and the phase separation is only detected at low salt concentration (below 500 mM NaCl). Although the purified RABV N0-P complex also phase separates, N protein is not required to observe LLPS in the acellular system. Taken together, all these data indicate that P plays a central role in biocondensate formation and is probably the core protein of NBs.

Interestingly, even in absence of N, P deletion mutants behave as in the minimal cellular system [7]. Particularly, PΔCTD is unable to induce phase separation. Thus, the role of P CTD in phase separation is probably unrelated to its N binding activity. Of note, PΔCTD, although unable to phase separate, is still recruited into P droplets. Here again, CTD deletion transforms P in a simple client of P droplets.

Aged droplets, when photobleached, do not recover their fluorescence. This indicates that they progressively undergo a transition to a gel which is their state at equilibrium [34]. Such a behavior has already been observed in other acellular minimal systems [35,36], less often in living cells that constitute systems that are out of equilibrium. However, in some cases, pathological transitions of liquid organelles to more viscous states [37] or solid amyloid-like aggregates [38] have been described and it has been shown that the cytoplasmic measles virus factories progressively harden with ageing [15].

## Several properties of droplets reconstituted in the acellular system are reminiscent of those of NBs

Several properties of P and N0-P droplets are relevant when considering NBs functions and evolution all along the viral cycle. First, P droplets concentrate RNA molecules, which may partially explain the sequestration of viral mRNAs in NBs [6] (but not the specificity of the process). Second, purified nucleocapsids, although binding P [23], are excluded from P condensates. Similarly, RNA molecules addition disrupts N0-P droplets which then lose their

sphericity. These phenomena are reminiscent of the ejection of RNPs from NBs [7]. Indeed, to be ejected, RNPs must undergo some conformational change or modifications of their physico-chemical properties which induce a decrease of their solubility inside the viral factory. It is plausible that *in vitro*, the RNP-like structures formed by the association of RNA with the N molecule are trapped in a state that is not soluble in P droplets. Finally, P droplets are able to associate with and of being enveloped by liposomes containing phosphatidylserine which is negatively charged. Again, this is reminiscent of NBs'association with a double membrane in the late stages of infection [6,18]. It is plausible that some positively charged patches of P exposed on the surface of the droplets drive the interaction with the liposomes. The huge number of P may then create an avidity phenomenon leading to a tight association between the droplet and the membrane which adapts its shape to that of the condensate. It is worth noting that interactions between liquid organelles and membrane-delimited ones have been shown to organize and reshape the latter [39]. A well characterized example is the ability of the synapsin liquid phase to capture small lipid vesicles and to induce their clustering [40–42].

## Final remarks

Using two distinct minimal systems, we have improved our understanding of the respective roles and dynamics of N and P in these two processes. Even if N is required in the cellular minimal system, RABV P appears to play a central role in LLPS. Globally, the *in vitro* data strongly suggest that the biocondensates formed by P (either in absence or in presence of N) have intrinsic physico-chemical properties from which the virus may take advantage. Obviously, in the cell, other viral or cellular proteins as well as post-translational modifications will facilitate some processes and make them more efficient, but the molecular bases of the underlying mechanisms are largely encoded in P. The ability to induce the formation of biocondensates recapitulating the properties of RABV factories will allow the characterization of the role of additional host and/or viral partners at the molecular level and may be useful in characterizing molecules modulating LLPS for future antiviral strategies [43].

## Material and methods

### Cell lines and virus

BSR cells, a clone from BHK 21 cells (Baby Hamster Kidney) were obtained from Dr A. Flamand (Former Laboratoire de Génétique des Virus, Gif-sur-Yvette, France). BSR-T7/5 cell line is a clone of BHK 21 constitutively expressing the T7 RNA polymerase [44]. HEK293T cells (human embryonic kidney, ATCC #CRL-3216) were purchased from ATCC organization (http://www.lgcstandards-atcc.org). BSR and HEK293T cells were grown in Dulbecco's modified eagle medium (DMEM) (Gibco-Life Technologies) supplemented with 10% fetal calf serum (FCS) (Gibco-Life Technologies), 100 units/mL of penicillin and 100 μg/mL of Streptomycin (AB 1X) (Gibco-Life Technologies). BSR-T7/5 cells were grown in the same medium supplemented with 1 mg/mL Geneticin (Gibco-Life Technologies).

Spodoptera frugiperda Sf21 and trichoplusia ni Hi-5 insect cells were cultured at 28˚C in TC-100 medium (Gibco) supplemented with 10% FCS, 100 units/mL of penicillin, 100 μg/mL of Streptomycin and 2.5 μg/mL amphotericin-β (Sigma).

The Challenge Virus Standard (CVS) strain of rabies virus was grown on BSR cells. Recombinant RABV rCVS-P-mCherry has been already described [26].

### Plasmids

The plasmids pTit encoding P and N proteins of CVS strain, referred as pTit-P and pTit-N, have been previously described [7]. In pTit plasmids [44,45], the gene of interest is under the

dependence of the T7 polymerase promoter and the corresponding transcripts contain an internal ribosomal entry site (IRES) located upstream the open reading frame. The plasmid pTit-P-mCherry has been previously described [7]. Plasmids pTit-GFP-N, pTit-N-106GFP107, pTit-N-112GFP113, pTit-N-128GFP129 and pTit-N-130GFP131 were constructed by inserting seamlessly the GFP sequence at indicated positions by Gibson Assembly method (NEB). For GFP insertion, GFP gene was flanked by two sequences encoding GGGGSGGGGS linkers. For GFP-N, a GGGGSGGGGS flexible linker was inserted between GFP and N. Plasmids pTit-N-103-*3L*-116, pTit-N-103-*1L*-108, pTit-N-107-*1L*-112, pTit-N-111-*1L*-116, pTit-373-*5L*-394, pTit-N-373-*2L*-382, pTit-N-381-*2L*-390 and pTit-N-385-*2L*-394 were constructed by replacing N residues by one to five stretches of GGGS motifs by Gibson Assembly method. Plasmid pTit-GFP-N-373-*5L*-394 was constructed by inserting GFP sequence at the amino-terminus of the pTit-N-373-*5L*-394 construct. A GGGGSGGGGS flexible linker was inserted between GFP and N. Plasmids pTit-N-R149A/R168A/R323A, pTit-N-R168A/R323A, pTit-N-R149A, pTit-N-R168A, pTit-N-R323A as well as pTit-GFP-N-R168A and pTit-GFP-N-R323A were constructed by changing arginine amino-acidic residues into alanines by site-directed mutagenesis. Plasmids pCDNA3.1-N, pCDNA3.1-N-103-3L-116, pCDNA3.1-N-373-5L-394, pCDNA3.1-P and pCDNA3.1-PΔNTD were constructed by subcloning the corresponding sequences from pTit plasmids.

Plasmid pET-22b+-PPV-C261S-StrepTagII encoding the full-length rabies virus P (Pasteur Vaccine, PV, strain) with a StrepTagII has been engineered by mutating C261 into a serine by site-directed mutagenesis. The plasmids pET-22b+-PPV-C261S-ΔNTD-StrepTagII and pET-22b+-PPV-ΔCTD-StrepTagII were constructed by means of Gibson Assembly by deleting amino-acids 1–52 and 195–296, respectively. In the pET-22b+-PPV-ΔCTD-StrepTag construct, the Cys297 residue belonging to the CTD domain has not been removed to label the protein with Cy3 maleimide.

The plasmids pFast-Bac-PPV and pFast-Bac-StrepTag-NPV have been used to transform DH10BAC bacteria (Invitrogen) and generate bacmids respectively encoding P and N from the PV strain. Those bacmids were then used to transfect Sf21 cells to generate the corresponding recombinant baculoviruses. Hi-5 cells were either infected by the baculovirus encoding P or co-infected with both baculoviruses to produce the N0-P complex.

## Antibodies and cell reactants

The rabbit polyclonal anti-P antibody was previously described [6] and was used at a 1:2000 dilution for immunofluorescence staining and at a 1:1000 dilution for western blot analysis. Monoclonal antibodies 26G6, 27A5 and 25C2 were previously described [46] and were used to immunoprecipitate P and PΔNTD. Mouse monoclonal anti-N antibody (81C4) was previously described [7] and was used at a 1:2500 dilution for immunofluorescence. Rabbit polyclonal anti-N (Institut Pasteur, Paris) was used for western blot and immunofluorescence. The mouse monoclonal anti-α-Tubulin (Sigma Aldrich) antibody was used at a 1:1000 dilution for western blot. Secondary fluorescent antibodies for immunofluorescence staining were purchased from Molecular Probes (Alexa fluor 488- or 568- conjugated). Secondary fluorescent antibodies for western blot analysis were purchased from Cell Signaling (Fluor 800-conjugated IgG or Fluor 680-conjugated IgG).

## Fluorescent probes and molecules

Cy5 poly-A RNAs of 10- and 40-nucleotides were purchased from Eurofins. Cy3 maleimide and NHS-Ester-Atto-488 were purchased from Lumiprobe and Sigma, respectively.

## Cell transfection

For all experiments, BSR-T7/5 cells were seeded at the density of 2.2 $10^5$ cells/well in a 12-wells plate. PEIMAX transporter 5 (polyscience) was used for all transfections following manufacturer's instructions. For simple transfection, 1 µg of total DNA was transfected. For double transfection, 400 ng of each plasmid were used.

For FRAP experiments, BSR-T7/5 cells were seeded at the density of 4.5 $10^5$ cells/dish (Ibidi). Two µg of DNA were used for simple transfections and 800 ng of each plasmid were used for double transfections.

## Immunoprecipitation

HEK293T cells (7.5 $10^5$ cells) were transiently co-transfected with pcDNA3.1 plasmids encoding N protein (either wild-type or mutants) and P protein (either full-length or PΔNTD) using Lipofectamine 2000 (1 µg DNA of each plasmid). Cells were harvested 24 h post-transfection and lysed in buffer containing 50 mM Tris-HCl pH7.4, 150 mM NaCl, 0.5% NP40, 1 mM NaF, 1 mM PMSF and a cocktail of protease inhibitors (#11-836-170-001 Roche Diagnostic). Samples were incubated for 30 min on ice and cleared by centrifugation (16,000 g, 10 min, 4˚C). Lysates were subjected to immunoprecipitation using a cocktail of anti-P monoclonal antibodies (26G6, 27A5 and 25C2) at 4˚C for 1 h before addition of magnetic protein A beads (Pierce) previously equilibrated with a buffer containing 50 mM Tris-HCl pH7.4, 150 mM NaCl and 0.05% NP40 for 1 h at 4˚C on a rotary wheel. Magnetic beads were washed three times with the same buffer. The beads were resuspended in Laemmli buffer prior to Western blot analysis.

## Western blot analysis

Cell extracts were prepared by lysis in 50 mM Tris-HCl pH7.4, 150 mM NaCl, 0.5% NP40, and a cocktail of protease inhibitors. Samples were incubated for 30 min on ice and cleared by centrifugation (16,000 g, 10 min, 4˚C). Proteins from cell extracts or immunoprecipitates were separated by electrophoresis on 12% SDS-PAGE and transferred onto a nitrocellulose membrane. The membrane was blocked with 5% skimmed milk in Tris buffered saline (TBS) for 30 min and incubated overnight at 4˚C with the corresponding antibodies. The blots were then washed extensively in TBS-0.1% Tween 20 and incubated for 1 h with Fluor-conjugated IgG secondary antibodies at room temperature (RT). After washing, the membranes were scanned with an Odyssey infrared imaging system (LI-COR, Lincoln, NE).

## Immunofluorescence staining and confocal microscopy

Cells were fixed for 15 min at RT with 4% paraformaldehyde (PFA), and further permeabilized for 5 min with 0.3% Triton X-100 in PBS. After saturation with 1% BSA in PBS, cells were incubated with the indicated primary antibodies for 1 h at RT, washed and incubated for 60 min with Alexa fluor conjugated secondary antibodies. Cells were then incubated for 10 min with 0.1 µg/mL DAPI in PBS before washing and mounting with Immu-mount (Thermo Scientific). Images were sequentially acquired using a laser scanning SP8 confocal microscope (Leica) with a 63x oil immersion objective (HC Plan Apo, NA: 1.4, Leica). We used 405, 488 and 561 nm wavelengths to excite DAPI, GFP and mCherry respectively and their fluorescence was collected through the following bandpass 415–455, 500–540 and 575–625 nm.

All images shown in the manuscript are representative of at least three independent experiments.

## Quantification of P-mCherry and GFP-N mutants concentrations in dense and dilute phases

To quantify the concentrations of P-mCherry and different fluorescent N mutants, we measured their areal fluorescence in both dilute (*i.e.* in the cytosol) and dense (*i.e.* in an N-P inclusion or in an NB) phases from confocal images. For this, we applied the method described in [25] by using Fiji/ImageJ software. We assumed that the local concentration of each protein is proportional to its local fluorescence intensity per pixel.

First, for each channel, the background intensity was defined by measuring the areal fluorescence intensity from a 50 by 50 pixels size box in the cytosol of an untransfected/uninfected cell. Then, a representative area of the dense phase was defined as a 7 by 7 pixels size box (in the rare cases of smaller inclusions, a 5 by 5 pixels box was used) in the middle of an inclusion. In the case of a cell containing several inclusions, the value of the areal fluorescence in the dense phase that we used in the graphs corresponded to the average of the areal fluorescence in each inclusion. A representative area of the dilute phase was also determined using a box of size 21 by 21 pixels close to the box used to measure the areal fluorescence of the dense phase. In order to determine the concentration of each protein in each phase, the background value was subtracted to the areal fluorescence in the dilute and dense phases.

For each condition, 30 cells were quantified from 3 independent experiments. For each independent experiment, we calculated the average areal fluorescence for each protein in each phase. For each condition, we compared the mean of the 3 experiments by Student t-test.

## Recombinant P proteins production in *E. coli* and purification

100 ml LBa (LB medium supplemented with 0.1 mg/ml ampicillin) was inoculated with a single colony of *E. coli* bacteria (Rosetta strain, Novagen) transformed with pET-22b+ plasmids encoding the different P constructs and cultivated overnight at 37˚C, 140 rpm. This preculture was diluted in 1L final of LBa and cultivated at 37˚C, 120 rpm. When the OD reached 0.8 AU, protein production was induced by addition of IPTG at the final concentration of 1 mM during 16 hours at 18˚C, 120 rpm. Bacteria were harvested and resuspended in 50 ml lysis buffer (20 mM Tris-HCl pH 7.4, 150 mM NaCl, 2 mM EDTA, 100 μg/ml lysozyme, and complete protease inhibitor cocktail) for 20 min on ice. The lysate was sonicated then centrifuged for 1 h, 4˚C, 20000 g. The cleared supernatant was loaded onto a gravimetric sepharose streptavidin column (Cytiva); P constructs were eluted in 20 mM Tris-HCl pH 7.4, 500 mM NaCl, 2 mM EDTA, and 5 mM desthiobiotine. Fractions containing P were pooled and concentrated using an ultrafree amicon concentrator (Millipore) with a cut-off of 30 kDa for the full-length and a cut-off of 10 kDa for the PΔNTD and PΔCTD deletion mutants. Purification was polished with a final gel filtration step using an S200 increase HR10-30 column (GE-Healthcare) in 20 mM Tris-HCl pH 7.4, 500 mM NaCl, 2 mM EDTA. After elution, the fractions containing P were pooled and concentrated up to 4 mg/ml and stored at -80˚C.

## P and N0-P complex production in insect cells and purification

RABV P (PV strain) was produced as described previously (21) with slight modifications. Hi-5 cells (Invitrogen) grown in Express 5 medium (Invitrogen) were infected with a baculovirus encoding P. Cells were harvested 60 hours post-infection and resuspended in 20 mM Tris-HCl pH 7.4, 150 mM NaCl, 2 mM EDTA, and complete protease inhibitor cocktail, before lysis by 3 cycles of successive incubations in liquid nitrogen and 37˚C. After centrifugation, the clear supernatant was loaded on a DEAE -Trisacryl M (Sigma) column, and P was eluted with a continuous 150 mM-300 mM NaCl gradient. Fractions containing P were pooled and precipitated

with 32% v/v saturated ammonium sulfate at 4˚C. P was recovered by centrifugation at 20,000 g at 4˚C for 1 hour and resuspended in 1 mL of buffer composed of 20 mM Tris-HCl pH 7.4, 150 mM NaCl, 2 mM EDTA, 2 mM β-mercaptoethanol. The protein was loaded onto a S200 HR10-30 (GE-Healthcare), fractions containing the pure P protein were pooled and concentrated to 1.5 mg/ml using an amicon ultrafree concentrator (Millipore) with a cutoff of 30 kDa. Protein was aliquoted and stored at -80˚C for further use.

The rabies virus N0-P complex (PV strain) was produced as previously described (24) with minor modifications. Hi-5 cells (Invitrogen) cultivated in Express 5 media (Invitrogen) were co-infected with baculovirus encoding P and N fused to a StrepTagII at its N-terminus. At 3 days post-infection, cells were harvested and resuspended (in 20 mM Tris-HCl pH 7.4, 150 mM NaCl, 2 mM EDTA and complete protease inhibitor cocktail) and lysed by 3 cycles of incubation in liquid nitrogen and 37˚C subsequently. After centrifugation, the clear supernatant was loaded on a gravimetric sepharose streptavidin column (Cytiva) and the fractions containing the N protein were eluted in 20 mM Tris-HCl pH 7.4, 500 mM NaCl, 2 mM EDTA, and 5 mM desthiobiotine. A final step of gel filtration using an S200 HR10-30 (GE-Healthcare) equilibrated in 20 mM Tris-HCl pH 7.4, 150 mm NaCl, and 2 mM EDTA allowed the separation of N0-P and N-RNA complexes. Fractions containing the pure N0-P complex were pooled and concentrated up to 1.5 mg/ml using a ultrafree amicon concentrator (Millipore) with a cut-off of 30 kDa. Protein was aliquoted and stored at -80˚C for further use.

## Viral nucleocapsids purification

Eighteen 150 cm$^2$ dishes of BSR cells were infected with RABV (PV strain) at an MOI of 0.1. Cells were harvested 3 days post-infection and resuspended in 2 mL of buffer containing 20 mM Tris-HCl pH 7.4, 150 mM NaCl, 2 mM EDTA, and supplemented by complete protease inhibitor cocktail, before lysis after dounce homogenization by 3 cycles of successive incubation in liquid nitrogen and 37˚C water bath. The supernatant was loaded on the top of a linear CsCl gradient (20–40% w/w in 20 mM Tris-HCl pH 7.4, 150 mM NaCl, and 2 mM EDTA buffer). The gradient was spun 16 h at 12˚C in a sw41 rotor at 40000 rpm. After ultracentrifugation, the visible band containing N-RNA complexes was punctured. The fraction was dialyzed against a 20 mM Tris-HCl pH 7.4, 200 mM NaCl, 2 mM EDTA buffer to get rid of the CsCl, and stored at -80˚C until further use.

## Protein labeling

P proteins were covalently labeled with Cy3-maleimide fluorophore (Lumiprobe). N0-P complex and nucleocapsids were labeled with Atto-488-NHS-Ester (Sigma). For labeling, probes and proteins were incubated 4 hours (probes/protein molar ratio 10:1) and labeled proteins were separated from the unliganded dye using a PD-10 (Merck Millipore) in a buffer containing 500 mM NaCl, 20 mM Tris-HCl pH 7.4, 2mM EDTA, and 2 mM β-mercaptoethanol.

## Experimental set-up for visualization of phase separation

Liquid droplets of P and N0-P were formed by mixing the different protein constructs in a buffer containing 20 mM Tris-HCl pH 7.4 and 2 mM EDTA. Molecular crowding agents (PEG 8000, Ficoll 400 and BSA) were used at the final concentration of 5% w/v unless otherwise specified. Non-fluorescent constructs (P, N0-P) or RABV nucleocapsids were mixed in the presence of 25–50 nM of fluorescent proteins. NaCl concentrations used ranged for 50 to 125 mM, unless otherwise specified. RNA-Cy5 was used at the final concentration of 100 nM. BSA-FITC was used at the final concentration of 250 μg/ml. Synthetic 40-nt poly-A RNA and total cellular RNAs extracted and purified from BSR cells (Nucleospin RNA, Macherey-Nagel) were used at 80 ng/μL to trigger P phase separation. Reactions were done in a 20 μL volume

and incubated at room temperature in 18-Well Ibidi Polymer wells (Ibidi). Micrographs were taken 1 hour after mixing the proteins unless otherwise specified.

## FRAP

In vitro, FRAP experiments were performed on droplets formed by non-fluorescent full-length P protein at 20 µM (containing 50 nM of fluorescent (Cy3) full-length P) in the presence of 5% PEG-8000 w/v, 20 mM Tris-HCl pH 7.4, 125 mM NaCl, 2 mM EDTA. FRAP on fluorescent RNAs was done in the same conditions with 100 nM of fluorescent RNAs. Alternatively, droplets formed by non-fluorescent N0-P complex at 2 µM in the presence of 5% PEG-8000 w/v, 20 mM Tris-HCl pH 7.4, 125 mM NaCl, and 25 nM of fluorescent (Atto-488) N0-P complex were photobleached. Droplets were formed and bleached less than 1 hour after demixing or more than 3–4 hours after demixing.

In cellulo, FRAP experiments were performed on BSR-T7/5 cells infected by RABV CVS (MOI = 0.5) and, immediately after a 1 h virus adsorption step, transfected with pTit plasmids encoding P-mCherry or GFP-N chimeras. FRAP was performed 14 h after infection/transfection. Alternatively, BSR-T7/5 cells were transfected for 24 h with pTit plasmids encoding P, N, P-mCherry and GFP-N chimeras (cellular minimal system). Cells were seeded in 35-mm glass-bottom µdishes (Ibidi) prior to transfection and/or infection.

Data acquisition was performed on an inverted Nikon Ti Eclipse E microscope coupled with a spinning disk (Yokogawa CSU-X1) and a 100x objective (Apochromat TIRF oil-immersion, NA: 1.49). Blue laser (488 nm, Vortran, 150 mW) was used for the excitation of fluorescent N0-P (Atto-488) in the acellular system, and GFP-N (as well as N-128GFP129 and N-112GFP113 chimeras) in transfected and infected cells. Yellow laser (561 nm, Coherent, 100 mW) was used for excitation of P protein (full-length) in the acellular system, and for P-mCherry in transfected and infected cells. Finally, a red laser (642 nm, Vortran, 100 mW) was used for the excitation of Cy5 RNAs concentrated into P droplets in the acellular minimal system. A quad-band dichroic mirror (405/491/561/641 nm, Chroma) and band-pass filters of 525/45 nm (Semrock), 607/36 nm (Semrock) and 692/40 nm (Semrock) were used respectively to ensure specific detection of the fluorescence. Images were recorded with a Prime 95B sCMOS camera (Photometrics). FRAP experiments were performed using iLas 2 module (GATACA Systems) and the whole system was driven by MetaMorph software version 7.7 (Molecular Devices).

FRAP experiments on cellular NB-like condensates were performed at the frame rate of 1 image every 2 seconds (resp. 1 image every second) for GFP-N chimeras (resp. P-mCherry), using the following sequence: 5 s of pre-bleach, 20 (resp. 40) ms to bleach GFP-N chimeras (resp. P-mCherry), and 120s post-bleach. Bleaching was performed in circular regions to bleach all of the condensate.

FRAP experiments on NBs in infected cells were performed at the frame rate of 1 image every 2 seconds (resp. 1 image every second) for GFP-N chimeras (resp. P-mCherry), using the following sequence: 5 s of pre-bleach, 20 (resp. 40) ms to bleach GFP-N chimeras (resp. P-mCherry), and 140 s (resp. 70 s) post-bleach for GFP-N chimeras (resp. P-mCherry). Bleaching was performed in circular regions to bleach either an entire NB or NB subvolumes.

FRAP experiments on droplets formed in the acellular minimal system were performed at frame rates of 2 images every second (for RNA-Cy5) to 1 image every 2 seconds (for P and N0-P). The following sequence was used to FRAP Cy3-P: 10 s of pre-bleach, 40 ms bleach, and 500 s post-bleach. The following sequence was used to FRAP Atto-488-N0-P: 10 s of pre-bleach, 20 ms bleach, and 300 s post-bleach. The following sequence was used to FRAP RNA-Cy5: 10 s of pre-bleach, 100 ms bleach, and 60 s post-bleach. All FRAP experiments were performed with lasers at 100% intensity.

For the FRAP data analysis we used a home-made macro on FIJI [47] performing the following steps. On every slide, the background was estimated and removed by measuring a region outside the cells. To correct for acquisitional bleaching, the mean fluorescence measured in the bleached region was normalized by the mean value of a region in a neighboring cell (if possible a condensate) or in a nearby droplet (for the acellular system). Then a second normalization was applied by dividing the results by the mean value of the pre-bleach sequence to allow comparison of recovery curves.

## Live microscopy

Droplets formation and fusion events made by full-length P at the final concentration of 10 to 20 µM (in 5% PEG 8000, 20 mM Tris-HCl pH 7.4, 125 mM NaCl, 2 mM EDTA) in the presence of 50 nM of Cy3-labelled protein were recorded by time-lapse epifluorescence microscopy at a frame rate of 1 image every 5 s. Data acquisition was performed on an AxioObserver epifluorescence microscope (Zeiss). Epifluorescence images were captured with a 63x oil immersion objective (Plan. Achromat, NA: 1.4, Zeiss). The laser wavelength for P-Cy3 excitation was 545nm.

## Association of P droplets with liposomes

For liposome preparation, we mixed 400 µg of phosphatidylcholine (type XVI-E from egg yolk, purchased from Sigma), 400 µg of phosphatidylethanolamine (type III from egg yolk purchased from Sigma), 200 µg of phosphatidylserine (from bovine brain, purchased from Sigma) and 2.5 µg of 2-(4,4-difluoro-5-methyl-4-bora-3a,4a-diaza-s-indacene-3-dodecanoyl)-1-hexadecanoyl-*sn*-glycero-3-phosphocholine (β-BODIPY 500/510 $C_{12}$-HPC) dissolved in organic solvents, and dried the mixture under vacuum. The lipid film was resuspended in 1 ml of buffer (150 mM NaCl and 5 mM Tris-HCl, pH 8). The liposome mixture was filter extruded and used immediately after preparation. For liposomes-P droplets experiments, liposomes were incubated at the final concentration of 100 µg/ml (~130 µM of lipids), with 20 µM of P protein in 5% PEG 8000, 20 mM Tris-HCl pH 7.4, 2 mM EDTA, and 125 mM NaCl.

## Differential interference contrast microscopy

Droplets formed by P in 5% PEG 8000, 20 mM Tris-HCl pH 7.4, 125 mM NaCl, 2 mM EDTA were observed on an inverted Nikon Ti Eclipse E microscope coupled with a Spinning Disk (Yokogawa CSU-X1) and a 100x objective (Apochromat TIRF oil-immersion, NA: 1.49).

## Calculation of P and N0-P droplets aspect ratios

Quantifications were performed using the image toolbox of MatLab software. Briefly, the grayscale image was converted to a binary image. The connected components (CC) of the output black and white image were then identified. For each CC, its eccentricity e (i.e., that of the ellipse that has the same second-moments as the CC) was calculated using MatLab software allowing the calculation of the aspect ratio a/b (where a and b are respectively the long and short axes of the ellipse) of each inclusion using the formula: $a/b = 1/(1-e^2)^{1/2}$.

## Statistical analysis

All presented experiments have been performed at least 3 times on independent samples. Results were analyzed by two-tailed student t-test or Kolmogorov-Smirnov test. The statistical significances (p-value) of the differences are indicated.

## Supporting information

**S1 Fig. Expression of chimeras between RABV N and GFP as well as N mutants. A)** and **B)** BSR-T7/5 cells were transfected for 24 h with pTit plasmids encoding the indicated fluorescent N constructions (GFP-N, N-106GFP107, N-112GFP113, N-128GFP129, N-130GFP131). **A)** Total expression of the indicated fluorescent N constructions was assessed by western-blot. GFP-N was revealed with a rabbit polyclonal anti-N antibody and tubulin was revealed with a mouse polyclonal anti-tubulin antibody followed by incubation with fluor-800 conjugated anti-rabbit and fluor-680 conjugated anti-mouse IgG. NT: non transfected cells. **B)** Cellular expression of the indicated fluorescent N constructions was assessed by fluorescence confocal microscopy on fixed cells. N was revealed with a mouse monoclonal anti-N antibody followed by incubation with goat anti-mouse IgG Alexa-488 antibody. **C)** and **D)** BSR-T7/5 cells were transfected for 24 h with pTit plasmids encoding the indicated mutants (mutated in loops L1 and L2). **C)** Total expression of the indicated mutants was assessed by western-blot. N was revealed with a rabbit polyclonal anti-N antibody (right panel) and tubulin was revealed with a mouse polyclonal anti-tubulin antibody (left panel) followed by incubation with fluor-800 conjugated anti-rabbit and fluor-680 conjugated anti-mouse IgG. NT: non transfected cells. **D)** Cellular expression of the indicated N mutants was assessed by fluorescence confocal microscopy on fixed cells. N was revealed with a rabbit polyclonal anti-N antibody followed by incubation with goat anti-rabbit IgG Alexa-488 antibody. **E-F)** HEK293T cells were transfected with the pCDNA3.1 plasmids coding the indicated constructs. Cells were lysed and complexes were immunoprecipitated with a cocktail of monoclonal antibodies. Cell extracts (inputs) and immune complexes (immunoprecipitates [IP]) were separated by SDS-PAGE and analyzed by Western blot using polyclonal anti-N and anti-P antibodies. **G)** BSR-T7/5 cells were transfected for 24 h with pTit plasmids encoding the indicated arginine mutants. Cellular expression of the indicated N mutants was assessed by fluorescence confocal microscopy on fixed cells. N was revealed with a mouse monoclonal anti-N antibody followed by incubation with goat anti-mouse IgG Alexa-488 antibody.
(TIF)

**S2 Fig. Characterization of P constructs. A)** SEC-MALS analysis of purified full-length P-Strep. **B)** Absorbance spectrum of purified P-Strep protein. **C)** SEC-MALS analysis of purified PΔNTD. **D)** SEC-MALS analysis of purified PΔCTD.
(TIF)

**S1 Movie. Sedimentation of P droplets (20μM containing 50 nM of fluorescent P) formed in presence of 3% PEG 8000, 125 mM NaCl and 20 mM Tris-HCl pH 7.4.**
(MP4)

**S2 Movie. Fusion events of P droplets formed in the same conditions as in S1 Movie.**
(MP4)

**S3 Movie. Fusion events of P droplets formed in the same conditions as in S1 Movie.**
(MP4)

**S1 Data. Raw data for Fig 1.**
(XLSX)

**S2 Data. Raw data for Fig 2.**
(XLSX)

**S3 Data. Raw data for Fig 3.**
(XLSX)

**S4 Data. Raw data for Fig 4.**
(XLSX)

**S5 Data. Raw data for Fig 6.**
(XLSX)

**S6 Data. Raw data for Fig 7.**
(XLSX)

## Acknowledgments

We are grateful to the core facilities at Institut Pasteur C2RT (Centre for Technological Ressources and Research) in particular to Bertrand Raynal (Molecular Biophysics). We thank Julien Sourimant for his careful reading of the revised manuscript.

## Author Contributions

**Conceptualization:** Quentin Nevers, Nathalie Scrima, Damien Glon, Romain Le Bars, Alice Decombe, Nathalie Garnier, Cécile Lagaudrière-Gesbert, Danielle Blondel, Aurélie Albertini, Yves Gaudin.

**Formal analysis:** Quentin Nevers, Nathalie Scrima, Damien Glon, Romain Le Bars, Alice Decombe, Nathalie Garnier, Cécile Lagaudrière-Gesbert, Danielle Blondel, Aurélie Albertini, Yves Gaudin.

**Funding acquisition:** Yves Gaudin.

**Investigation:** Quentin Nevers, Nathalie Scrima, Damien Glon, Alice Decombe, Nathalie Garnier, Malika Ouldali, Danielle Blondel, Aurélie Albertini, Yves Gaudin.

**Methodology:** Quentin Nevers, Nathalie Scrima, Damien Glon, Romain Le Bars, Alice Decombe, Nathalie Garnier, Malika Ouldali, Danielle Blondel, Aurélie Albertini, Yves Gaudin.

**Project administration:** Yves Gaudin.

**Resources:** Yves Gaudin.

**Software:** Romain Le Bars.

**Supervision:** Nathalie Scrima, Aurélie Albertini, Yves Gaudin.

**Validation:** Quentin Nevers, Aurélie Albertini, Yves Gaudin.

**Visualization:** Quentin Nevers, Nathalie Scrima, Malika Ouldali, Yves Gaudin.

**Writing – original draft:** Yves Gaudin.

**Writing – review & editing:** Quentin Nevers, Nathalie Scrima, Damien Glon, Romain Le Bars, Cécile Lagaudrière-Gesbert, Danielle Blondel, Aurélie Albertini, Yves Gaudin.

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
