## [Decision Letter · Decision Letter 0]

1 Jun 2022

Dear Dr. Gaudin,

Thank you very much for submitting your manuscript "Properties of rabies virus phosphoprotein and nucleoprotein biocondensates formed in vitro and in cellulo" for consideration at PLOS Pathogens. As with all papers reviewed by the journal, your manuscript was reviewed by members of the editorial board and by several independent reviewers. In light of the reviews (below this email), we would like to invite the resubmission of a significantly-revised version that takes into account the reviewers' comments.

The manuscript has now been seen by four reviewers and was found to provide sufficiently novel insights into the molecular mechanisms of RABV condensate formation and function.

However, a fairly large number of major issues were raised, including:

- Specific concerns with FRAP experiments including their representation and interpretation

- The difference of NBs obtained with or without infection are not discussed in the text. The authors should calculate the Cdense/Cdiffuse as described the Brangwynne group (Riback et al. Composition-dependent thermodynamics of intracellular phase separation. Nature 581, 209–214 (2020).

- The relevance of data obtained with E.coli-expressed P-protein needs to be confirmed as RABV P-protein does not phase-separate when expressed in cellula but E.coli-expressed purified P-protein phase-separates in vitro. Will purified P-protein expressed in eucaryotic cells also condensate in vitro?

- RNA recruitment into condensates needs to be assessed and possible RNA contamination needs to be ruled out.

Authors should address all the points raised by the reviewers.

We cannot make any decision about publication until we have seen the revised manuscript and your response to the reviewers' comments. Your revised manuscript is also likely to be sent to reviewers for further evaluation.

Sincerely,

Ralf Altmeyer

Guest Editor

PLOS Pathogens

Urs Greber

Section Editor

PLOS Pathogens

Kasturi Haldar

Editor-in-Chief

PLOS Pathogens

orcid.org/0000-0001-5065-158X

Michael Malim

Editor-in-Chief

PLOS Pathogens

orcid.org/0000-0002-7699-2064

The manuscript has now been seen by four reviewers and was found to provide sufficiently novel insights into the molecular mechanisms of RABV condensate formation and function.

However, a fairly large number of major issues were raised, including:

- Specific concerns with FRAP experiments including their representation and interpretation

- The difference of NBs obtained with or without infection are not discussed in the text. The authors should calculate the Cdense/Cdiffuse as described the Brangwynne group (Riback et al. Composition-dependent thermodynamics of intracellular phase separation. Nature 581, 209–214 (2020).

- The relevance of data obtained with E.coli-expressed P-protein needs to be confirmed as RABV P-protein does not phase-separate when expressed in cellula but E.coli-expressed purified P-protein phase-separates in vitro. Will purified P-protein expressed in eucaryotic cells also condensate in vitro?

- RNA recruitment into condensates needs to be assessed and possible RNA contamination needs to be ruled out.

Authors should address all the points raised by the reviewers.

Reviewer's Responses to Questions

**Part I - Summary**

Reviewer #1: In this article the authors examined the impact of N mutations on the biogenesis of biocondensates in cells and described various properties of biocondensates formed in vitro by P alone or N0-P. Numerous exciting observations are presented in these different models providing insights into the biogenesis and evolution of Negri bodies during infection. However, these issues are addressed only superficially and a systematic characterization is lacking to strongly support the results and the proposed interpretations.

For the first part, which concerns the study of biocondensates formed in cells upon coexpression of N and P, the authors identify certain residues/regions as important to drive LLPS. For some of these modifications, the authors hypothesize that the ability to induce biocondensate formation depends on binding to RNA or P but do not test these hypotheses.

The second part studies the properties of biocondensates formed in vitro from P alone and N0-P complexes. These objects could be of interest as models even if such condensates do not exist in cells. The authors tested the effect of NaCl concentration, RNA addition or liposomes. The experiments are limited to restricted experimental conditions without detailed explorations. The experiments performed on N0-P condensates are also preliminary. They lead to observations that are mechanistically appealing but should be explored in more depth.

Parts 1 and 2 are independent and are not linked.

Reviewer #2: The role of condensates in RNA virus replication is a highly active research area that is of interest to virologists, physical biologists and drug discovery professionals. Rabies is an important and interesting virus. Thus, the area covered by this paper is of high topical interest.

A role of liquid condensates, called Negri bodies, in Rabies virus replication was already published. That prior work is appropriately cited in the submission and the authors are careful to distinguish between confirmatory and new findings.

The experimental work is well executed and analyzed throughout.

Figs 1 reports and characterize a minimal system comprising P and N proteins that is sufficient to assemble into condensates in living cells. This system is useful for probing the biophysics of replication condensate assembly and might find use in drug discovery. As such, it is a significant step forward. An interesting feature is that the P protein turns over rapidly within these condensates while the N protein does not. This is an unusual finding. Its mechanistic basis is not clear, though mutagenesis of N takes some early steps towards provide molecular-level understanding.

Fig 2 investigates condensates in RABV infected cells and confirms the interesting finding that P is highly dynamic while N is not. This figure is in part confirmatory of previous work, but the protein dynamics data are novel and the work is important to provide physiological relevance to the minimal systemin Figure 1.

Fig 3-6 report and characterize a series of pure protein experiments in which P, alone or together with N and RNA, undergoes condensation and phase separation in vitro. Crowding agents are necessary, a positive is that the authors show the condensates assemble with multiple crowing agents. in the presence of crowding agents. I have mixed feelings about these experiments. On the positive side, it is useful and interesting to have a pure protein reconstitution side to the submission. This is highly complementary to the experiments in living cells and sets the stage for more detailed biochemistry and perhaps drug discovery. On the negative, use of crowding agents such as 5% PEG to induce phase separation has been criticized as potentially unphysiological. Use of crowing agents was common in the early years of the condensate field, but it led to the sense that condensation under artificial crowing conditions, which may be much more crowded than the interior of cells, can be an artifact. The debate over use of crowding agents continues and it is important to be cautious in interpreting experiments that use crowding agents to promote condensation of pure proteins. One control is to use mutants to test whether the molecular requirements for condensate assembly are similar in living cells and a pure protein + PEG system. The submission does a pretty good job at that, which makes the data collected with PEG convincing in terms of in vivo relevance. Given the validation by mutant analysis, mechanistic conclusions can be drawn from the PEG-crowded system, though caution will always re required in interpretation.

One specific concern with Figs 3-6 is that hardening of protein condensates over time can be caused by RNA contamination. Its important that the authors provide documentation that their protein preparations are free of contaminating RNA, which often co=purifies with RNA binding proteins and can be challenging to remove.

Overall, the paper provides significant new information about how RABV P and N proteins assemble intro condensates and the different dynamics of P and N in these condensates. The concepts are not new, the details are new, the experiments are well done. Use of mutants ot validate minimal and reconstituted approaches is an important positive of the submission. The systems reported might be useful for anti-viral drug discovery, which is another significant positiv. Overall, the work is clearly worth publishing, To this reviewer, the paper is somewhat lacking in information relevant to understanding the precise molecular interfaces involved in condensate assemble and the precise role of condensate biophysics in virus infection. These deficiencies, combined with a lack of conceptual novelty, decrease impact, but it is still a useful paper on an important virus.

Reviewer #3: The laboratory of Yves Gaudin had previously shown that Negri bodies, the viral factories specialized for transcription and replication of Rabies virus (RABV), have liquid properties, and are formed by liquid-liquid phase separation (LLPS). They also showed before that the co-expression of RABV nucleoprotein (N) and phosphoprotein (P) in mammalian cells is sufficient to induce the formation of cytoplasmic biomolecular condensates having properties that are like those of NBs. In addition, they showed, using a cellular minimal system which domains in P are essential for the formation of biomolecular condensates. Now they further characterized the mechanisms of RABV P- and N-induced LLPS. Here they describe the characteristics of N- enabling LLPS and the role of P in the formation of biomolecular condensates. They also highlight the physicochemical features of P and N0- P that may explain some of NBs properties and function. The manuscript is very well written, organized, and the study is robust. Nevertheless, this reviewer points to the lack of quantification and measurements that could elucidate the thermodynamics associated to this process, namely the conditions that lead to higher concentration of material inside NBs or in the diffuse phase. These are important measurements that would increase the contribution of the manuscript to the field of phase separation.

Reviewer #4: I think that this paper is very nice with many ideas about rabies and all negatve strand RNA viruses. This paper is written by the people that first seen that Negri bodes are "droplets"! Now, they show how the nucleoprotein bind to P to making the droplets. Also, very new, they add negative charged lipids and they form a membrane around the droplet, just what happens in the cell, before making new viruses

**Part II – Major Issues: Key Experiments Required for Acceptance**

Reviewer #1: 1. How many times were each of the experiments repeated? The number of experiments performed is never indicated in the figure legends except for fig2d.

2. Many FRAP experiments are presented and raise several questions.

a. The FRAP curves presented are averages of a limited number of data points, almost always less than 10. Some curves are from les than 5 data points (Fig1c 4 or 5 points, Fig3h 3 points, Fig5b 4 or 5 points, fig6e 3 points). Moreover, the number of independent experiments done for collecting these data is never indicated.

b. Some FRAP curves are not shown: Fig1c: FRAP curves for the N-128GFP129 mutant and Fig2e: FRAP curves for subvolumes

c. Fig1c: For the N-128GFP129 mutant, on the images shown the small condensates change suggesting a change of plane during the acquisition which would impede the interpretation of the results. How is this controlled?

d. The results of the FRAP experiments seem sometimes in contradiction with the images :

-Fig5b, panel 10nt >4h. The signal at 50s seems equivalent to the pre-bleach signal whereas this is not the case on the curves

-Fig6f. The curves do not correspond to the images shown. One observes here an important recovery, similar to that observed on the fig3h whole droplet panel <1h, whereas the curves are very different. The conclusion (line 247) "fluorescence recovery is an extremely slow process" seems not to be poorly supported by the images. The curves are only from 3 acquisitions.

-Fig5b. Overall in the partial bleaches, it seems that the fluorescence of the whole structure is strongly decreased after the bleach. This aspect is not commented nor taken into account in the analyses.

e. Fig5b: Why the time scales do not go further for RNAs in particular for the condensates examined after 4 h. The absence of recovery at 50s does not allow to conclude that recovery is abrogated (line 285). The same type of fluorescence redistribution kinetics is observed for P (fig3h <1h).

3. In the experiments on cells with or without condensate formation, no quantification is provided. The number of cells analyzed and the number of experiments should be mentioned Fig1b Fig1e Fig1g Fig2a, b and c

4. Do the N mutants support transcription-replication activity and inhibit transcription-replication in the presence of wild-type N? Are they incorporated into virions?

5. Lines 228-241: A complete description of the in vitro P condensation model and its sensitivity to salinity requires phase diagrams

6. To rigorously describe LLPS inhibition by RNAs, a phase diagram is required.

7. Lines 308-313 (fig 6h) Mixtures of P and NC could lead to observations of interest but only an image without quantification is presented.

8. Lines 314-318: Is RNA recruited to the droplets? Can the concentration of RNA disassemble the condensates as before? Are the results of the shape analysis different when RNAs are added before or at the same time as the crowding agent?

Reviewer #2: I would like to see OD280/260 ratios for the purified proteins to demonstrate that RNA contamination is not an issue inn the pure protein experiments. If this is in the methods and I missed it then apologies

Reviewer #3: This reviewer highlights some small points to address:

1- Figure 1 – The NB formed when co-expressing of N and P without infection are different in number and size relative to infection. The properties may be similar, but the characteristics of these NBs seem different relative to the concentration of material inside NBs and diffuse in the cytosol, which underlies the ability of material to phase separate. For example, in the controls GFP-N and P of Figures 1B, 1E and 1N, there is a relative high diffuse phase relative to concentrated in NBs for N, as compared to the low levels of diffuse N in infected cells at 14 h. The reasons for the differences are important to assess and absent in the manuscript. Would this be a matter of concentration? The authors could test the levels of material by western blot in both conditions (co-expression and infection/transfection), and assess Cdense/Cdiffuse by quantifying mean intensity, as in the manuscript "Composition-dependent thermodynamics of intracellular phase separation"

https://www.nature.com/articles/s41586-020-2256-2.pdf and test whether varying the concentration of either N, P and N and P results in different concentration of material inside and outside NBs.

2- By calculating Cdense and Cdiffuse as suggested above and by integrating the data with the western blots of S1C Fig and others, the authors would provide robust measurements taking into account cellular variation rather than stating in lines 180-181 “appeared to be higher than with wild-type N protein (as judged by the intensity of the cytoplasmic fluorescence intensity)”.

3- In figure 2B, the measurements of the concentration of N in the diffuse phase versus the dense phase would make the results robust and reveal biological properties of the condensates.

4- P ability to undergo phase separation is different in cells or in P purified from E. coli -P purified from E. coli phase separates while in cells, P does not phase separate on its own. However, it is unknown whether P purified from cells can undergo phase separation. This distinction is important, as for example, it may suggest that P may contain post-translational modifications that abrogate phase separation. The purification of P from cells and associated studies on its ability to undergo phase separation are key to elucidate this point.

Reviewer #4: I did not find any problems

**Part III – Minor Issues: Editorial and Data Presentation Modifications**

Reviewer #1: Minor points

1. Specify the FRAP protocol: (line 633) photobleaching is controlled fluorescence measurement in a neighboring cell. How is this region defined and is it a condensate? How is it corrected in vitro?

2. Fig1b, 1e and 1g: The condensates mentioned in the text are barely visible (especially in 1g). For Fig1b, a wild-type N + P control would be relevant.

3. Fig2d: How were the percentages of infected and transfected cells obtained? How many cells were analyzed for each experiment?

4. Hypotheses should be proposed or explored to explain the dominant negative effect of N-106GFP107 even though it is not recruited to NBs suggesting that it is not incorporated into the RNP.

5. Lines 245-247: The redistribution kinetics of P fluorescence are very different in cells (fig1c) and in droplets in vitro (fig3h) (even fresh). This point should be discussed.

6. For experiments mixing molecules with P or N0-P condensates, the molar ratio of the molecules brought together RNA versus P; lipids versus P, RNA versus N0-P, NC versus P should be given.

7. An association between liposomes and P condensates is observed. Is it sensitive to the concentration of liposomes? How does it evolve over time?

8. Line 279, Fig5a: The RNA concentration (Fig5a) is not mentioned. The time of RNA addition is not specified (addition before droplet formation or on already formed droplets)

9. The units of RNA concentrations for Fig5b are in nM whereas they are in ng/µL in Fig5c.

Reviewer #2: Perhaps the discussion could be sharpened a little to focus on new mechanistic findings and their relevance to viral replication

Reviewer #3: Figure 2 – The levels of infected and transfected cells are very low. In the methods, the authors state that they infect and transfect 1h after infcetion. This may already activate an antiviral state in the infected cell. Have the authors attempted to infect and transfect at the same time to improve the efficiency of infected and transfected cells?

Reviewer #4: (No Response)

PLOS authors have the option to publish the peer review history of their article (what does this mean?). If published, this will include your full peer review and any attached files.

Reviewer #1: No

Reviewer #2: No

Reviewer #3: No

Reviewer #4: No
---

## [Decision Letter · Decision Letter 1]

23 Nov 2022

Dear Mr Gaudin,

We are pleased to inform you that your manuscript 'Properties of rabies virus phosphoprotein and nucleoprotein biocondensates formed in vitro and in cellulo' has been provisionally accepted for publication in PLOS Pathogens.

Best regards,

Urs F. Greber

Section Editor

PLOS Pathogens

Kasturi Haldar

Editor-in-Chief

PLOS Pathogens

orcid.org/0000-0001-5065-158X

Michael Malim

Editor-in-Chief

PLOS Pathogens

orcid.org/0000-0002-7699-2064

Reviewer Comments (if any, and for reference):

Reviewer's Responses to Questions

**Part I - Summary**

Reviewer #1: I thank the authors for answering my questions and making improvements to the manuscript. I agree with the authors on the interest of their results but regret that some aspects are just touched upon. However, the added data have strengthened the manuscript. The analyses of P and N concentrations in the dense and dilute phases are particularly interesting.

**Part II – Major Issues: Key Experiments Required for Acceptance**

Reviewer #1: Discussion point: The authors point out that the sections on in-cell N and in vitro P allow a comparison of the systems. This comparison is not clearly apparent in the manuscript. The differences in P mobilities in the 2 systems as assessed by FRAP should be clearly discussed, which is not the case.

Important points remain to be clarified

1. Sup Fig 1D and 1E: N373-5L-394 and N385-2L-394 mutants (to a lesser extent) have decreased MW while the theoretical MW is only slightly changed (20AA replaced by a 20AA linker). Is there a cleavage? To avoid potential bias, a co-precipitation study of interaction-N should be presented using the 373-2L-382or 381-2L-390 mutants whose sizes are consistent with predictions.

2. As pointed out previously the images presented on Fig7F are not convincing: the bleach is partial, the condensate is very small, the recovery seems much larger than for the average. It is true that there is a variability between replicates but a more representative image should be presented. I would like to see a few other images.

3. Fig 7 comments (L360…) Some elements are missing to conclude that the N0-P "condensates" are formed by LLPS especially as they are less spherical and the mobility of the complex is very low. It would be necessary to have fusion images of the condensates to support this assertion.

4. I must disagree with the authors who provide the approximate number of experiments performed in the methods (at least 3 experiments performed). For the reader to immediately and easily assess the robustness of the data, specific information should be provided in the legends: how many experiments are represented when representative data from one experiment are shown (Images), exact number of experiments (not a range) and data points, names of the statistical tests (which are given here). This is done for the Fig 3.

5. I agree that conclusions drawn from FRAP data are qualitative and that their interpretation is cautious. I still consider that 5 points is a very small number especially coming from 3 independent experiments. Why performing FRAP on only one structure in an experiment ? (If there are technical issue, they should be explained)

6. I thank the authors for their response regarding the “correction” of the movements of condensates in x, y and z. This information is to be included in the methods since it describes how data are produced.

“There are sometimes movements of droplets. It is unavoidable. For this reason, we coded a second macro, which allowed to correct the movement of the condensates (and also of the NBs for that matter) in X and in Y. FRAP curves for which there was an obvious movement of the condensates on the Z axis were discarded from the analysis. Furthermore, when we did not observe recovery, we controlled the absence of an inclusion in another z plane near the bleached zone.”

**Part III – Minor Issues: Editorial and Data Presentation Modifications**

Reviewer #1: 1. The additional data provided below could be of interest and could be added in supplementary information (?)

“We showed in the images presented in the previous figure 6I (now 7J) that we had a similar disruptive effect regardless of RNA addition times (10 minutes before PEG addition, at the time of PEG addition or 10 minutes after PEG addition). We had quantified this effect in each condition. However, we had decided to present only the effect observed in the latter condition (addition of RNA 10 minutes after PEG addition) for better readability of the figure. Below is the full figure, the reviewer will see that, unsurprisingly, the disruptive effect of RNA is detected earlier when RNA is added at early times. + Figure”

2. Regarding the indication of the number of analyzed cells on fig 1b, 1e and 1g, the fact that all the cells present the exact same phenotype on transfection is of interest. The number of transfected cells observed and registered should be mentioned.

3. Line 172, fig S1: Could the authors clarify why L2 does not allow interaction with P of the N103- 3L-116 mutant?

4. My question “Do the N mutants support transcription-replication activity and inhibit transcription-replication in the presence of wild-type N?” was not clear. I meant transcription-replication activity in a minireplicon system when plasmids are co-transfected. In such conditions, most of the cells are expressing all the proteins.

5. Previous minor point 3. Fig2d: How were the percentages of infected and transfected cells obtained? How many cells were analyzed for each experiment?

6. My question was unclear. I wanted to know how you did count the infected or transfected cells: naked eyes or automatically (if automatically how?)

7. Fig7F: FRAP data are difficult to interpret. The 2 proteins are labeled while they do not necessarily have the same behavior. This point should be emphasized.

PLOS authors have the option to publish the peer review history of their article (what does this mean?). If published, this will include your full peer review and any attached files.

Reviewer #1: No

---

## [Editor Report · Acceptance letter]

2 Dec 2022

Dear Mr Gaudin,

We are delighted to inform you that your manuscript, "Properties of rabies virus phosphoprotein and nucleoprotein biocondensates formed in vitro and in cellulo," has been formally accepted for publication in PLOS Pathogens.

Best regards,

Kasturi Haldar

Editor-in-Chief

PLOS Pathogens

orcid.org/0000-0001-5065-158X

Michael Malim

Editor-in-Chief

PLOS Pathogens

orcid.org/0000-0002-7699-2064